# More than 17,000 tree species are at risk from rapid global change

Coline C. F. Boonman ●[1] ✉, Josep M. Serra-Diaz[2,3], Selwyn Hoeks ●[4], Wen-Yong Guo[5,6], Brian J. Enquist[7], Brian Maitner ●[8], Yadvinder Malhi ●[9,10], Cory Merow ●[2], Robert Buitenwerf ●[1] & Jens-Christian Svenning ●[1]

Trees are pivotal to global biodiversity and nature's contributions to people, yet accelerating global changes threaten global tree diversity, making accurate species extinction risk assessments necessary. To identify species that require expert-based re-evaluation, we assess exposure to change in six anthropogenic threats over the last two decades for 32,090 tree species. We estimated that over half (54.2%) of the assessed species have been exposed to increasing threats. Only 8.7% of these species are considered threatened by the IUCN Red List, whereas they include more than half of the Data Deficient species (57.8%). These findings suggest a substantial underestimation of threats and associated extinction risk for tree species in current assessments. We also map hotspots of tree species exposed to rapidly changing threats around the world. Our data-driven approach can strengthen the efforts going into expert-based IUCN Red List assessments by facilitating prioritization among species for re-evaluation, allowing for more efficient conservation efforts.

Earth's biosphere functioning is highly dependent on trees, which are essential ecosystem engineers[1,2] and generate habitat to half the world's known terrestrial flora and fauna[3,4]. Furthermore, tree diversity holds significant cultural and spiritual value, provides economically valuable products for national and global trade[5,6], and enriches local livelihoods and societal fabric. Despite their undeniable importance, the rapid intensification and expansion of human activities[7] during the Anthropocene poses severe threats to trees, driving habitat loss, fragmentation, degradation, and over-exploitation that could culminate in the extinction or decline of many tree species[5,7,8]. Such declines and losses would trigger profound repercussions across all trophic levels[2,7,9]. Therefore, an intensive, targeted approach to tree conservation is critical to prevent species extinctions. An important initial

step involves assessing current threats towards each tree species[5,10]. Given the rapid rise in relevant data sources[11–15], a data-driven approach has a potential to facilitate the formulation of targeted and effective strategies to alleviate the impending risk of extinction.

Changing spatial patterns in threats (threat landscapes) are the primary cause for changes in extinction risk[16]. However, existing automated conservation status assessment approaches do not typically include information on recent temporal changes in threat exposure and intensity[8,17–22]. For example, when such temporal changes are considered, extinction risks for Chinese woody species are projected to increase by more than 50% by 2070[8], highlighting the importance of changes in threat landscapes. Temporal threat dynamics, even of those with less easily diagnosable impacts like climate change, provide

[1]Center for Ecological Dynamics in a Novel Biosphere (ECONOVO) & Center for Biodiversity Dynamics in a Changing World (BIOCHANGE), Department of Biology, Aarhus University, Aarhus, Denmark. [2]Department of Ecology and Evolution and Eversource Energy Center, University of Connecticut, Storrs, CT, USA. [3]Université de Lorraine, AgroParisTech, INRAE, Silva, Nancy, France. [4]Department of Environmental Science, Radboud Institute for Biological and Environmental Sciences (RIBES), Radboud University, Nijmegen, The Netherlands. [5]Research Center for Global Change and Complex Ecosystems, School of Ecological and Environmental Sciences, East China Normal University, Shanghai 200241, People's Republic of China. [6]Zhejiang Tiantong Forest Ecosystem National Observation and Research Station, School of Ecological and Environmental Sciences, East China Normal University, Shanghai 200241, People's Republic of China. [7]Department of Ecology and Evolutionary Biology, University of Arizona, Tucson AZ 85721, USA. [8]Department of Geography, University at Buffalo, Buffalo, NY, USA. [9]Environmental Change Institute, School of Geography and the Environment, University of Oxford, South Parks Road, Oxford OX1 3QY England, UK. [10]Leverhulme Centre for Nature Recovery, University of Oxford, Oxford, UK. ✉e-mail: colineboonman@bio.au.dk

complementary information to experts in extinction risk evaluations and can ultimately help achieving more efficient, comprehensive, timely, and targeted conservation and restoration effort responses.

With the latest Global Tree Assessment (GTA), 92.7% of all 57,922 tree species[23] have been assigned a conservation status that will be included on the International Union for the Conservation of Nature (IUCN) Red List of Threatened Species[5,24] (hereafter 'IUCN Red List'). This list provides valuable, expert-validated, species-specific extinction risk assessments and catalyzes biodiversity conservation action and policy change[25,26]. The GTA identified nine threats to tree species: agricultural expansion (affecting 29% of all tree species), over-exploitation (27%), livestock farming (14%), urban development (13%), fire regime changes (13%), energy production and mining (9%), wood and pulp plantations (6%), the spread of invasive and other problematic species (5%), and climate change (4%)[5,27]. However, 7,700 tree species are labeled as Data Deficient by the IUCN, representing more than 13% of all tree species. Further, while impacts of some threats such as deforestation may be relatively easy to detect, impacts of other threats are less straightforward to diagnose and risk may be overlooked. Notably, impacts of climate change may be difficult to determine as they may not only become realized as direct effects, e.g., increased drought frequency and severity resulting in higher mortality in adult trees and lower recruitment, but also via indirect mechanisms, e.g., where a tree species loses its main animal dispersal agent(s)[28] or simply lacks the dispersal ability required to track its suitable climate conditions[29]. In addition, the prevalence and intensity of all threats vary strongly in time and space, and threats may overlap with unknown synergistic effects, adding to the complexity of estimating their impact consistently across species[16,30]. Together with the need for laborious expert-based IUCN Red List re-evaluations every five to ten years[31], the risks of overlooking pressures especially for rare species in remote areas, suggest that a data-driven systematic approach that quantifies recent changes in threats could aid experts in prioritizing species for new in-depth conservation assessments or re-evaluations[32] and therefore could strengthen the critical IUCN Red List assessment work[27,30–34].

Applying a data-driven, species-specific strategy, we aimed to enhance our understanding of the magnitude of global changes to which tree species are exposed. First, we quantified the rates of change of the estimated extent of each tree species over the past two decades for six of the major threats to trees, as identified by the GTA[5]: (I) crop agriculture expansion, (II) tree cover decline as a proxy for overexploitation in all vegetation types, (III) urban built-up area expansion, (IV) deforestation as a proxy for land-use change threats in all forested areas, (V) changes in burned area as a descriptor for fire and fire suppression, and (VI) climate change as measured by changes in annual values of minimum and maximum temperature, vapor pressure deficit (VPD) and VPD seasonality, precipitation, and precipitation seasonality. Our approach utilizes species' extent, determined as the minimum convex polygon encompassing 95% of species' high-quality occurrence records, with areas of unsuitable climate and water bodies removed, allowing us to calculate the exposure to threats independent of each species' sensitivity or adaptability to individual threats. In essence, we refined a tree species' extent of occurrence with broadly defined climate niches to remove often vast areas with clearly unsuitable conditions while suggesting that the species potentially has unregistered occurrences in areas currently designated as unoccupied. In particular, the first four threats - all related to land-use change - pose extinction risks to any tree species, regardless of its specific characteristics. Next, we evaluated the congruence between rates of recent change and the different IUCN conservation status categories. We then propose a prioritization for expert-based IUCN Red List re-evaluation based on high exposure to recent changes in threats. Lastly, we mapped the distribution of these high-priority candidates to identify hotspots of tree species exposed to high rates of recent global change. This identification can direct conservation efforts and data gathering, thereby aiding future expert assessment and conservation efforts.

Our study presents a data-driven approach focusing on recent changes in anthropogenic threats to inform conservation assessments. We identify a substantial underestimation of threats faced by tree species globally. Out of the 41,835 species included in this study (72.2% of all tree species worldwide), we were able to calculate rates of recent threat exposure for 32,090 species, of which 54.2% (17,393 species) were exposed to major and increasing threats. While these changes in exposure are likely to increase tree species' extinction risk, only 8.7% of these species were listed as threatened on the IUCN Red List. This shows that current IUCN Red List statuses do not adequately capture changes in exposure to threats. Additionally, the extent of 9,741 tree species were too small to calculate rates of recent change in threat exposure. However, we estimate these species on average to be exposed to higher rates of change for urban area expansion, tree cover decline, deforestation, and for the climate change component VPD. Our data-driven method, applicable to any taxa, can help expedite IUCN Red List assessments, facilitating timely, efficient development of species-specific conservation strategies. Additionally, it aids in identifying hotspots of species exposed to rapid global change, thus informing decision-making for conservation area allocation.

## Results
### Exposure of tree species to threats
We described the exposure to six significant threats for 32,090 tree species by quantifying rates of recent change within and relative to species' extent (Fig. 1). The largest changes in potential threats within species' extent were due to deforestation, which only considers the reduction of tree cover to zero in previously forested (more than 50% tree cover) locations, and tree cover decline, which includes any amount of tree cover reduction in any given location. Tree cover decline occurred at a median rate of 1.86% of species' extent per year (3.55% as 95th quantile) and deforestation at a median rate of 0.41% of species' extent per year (1.85% as 95th quantile; Supplementary Table 1). Accumulating across the years, maximum change even went up to 100%. More specifically, the maximum annual tree cover decline was 6.67% of species' extent, and was found for *Rhodolaena macrocarpa* (native to Madagascar[35], Red List status: Endangered due to wood harvesting, mining, and fire), *Arytera miniata* and *Syzygium sambogense* (both native to New Guinea[35], Red List status: Endangered due to deforestation for settlement and agriculture activity, and Not Evaluated, respectively), *Gluema korupensis* (native to Cameroon[35], Red List status: Endangered due to deforestation and its low number of mature individuals), and *Weberbauerocereus madidiensis* (native to Bolivia[35], Not Evaluated on the IUCN Red List). Additionally, the maximum annual deforestation rate was 5.0% of species' extent, and was found for *Eucalyptus redimiculifera* (native to Western Australia[35]; Red List status: Data Deficient). These comparisons of species exposed to the fastest rates of change to their expert-based assessments from the IUCN Red List show that the species most exposed to tree cover declines are indeed listed as Endangered due to wood harvesting on the IUCN Red List. These, and other comparisons below, support our method and suggest that species without a Red List status or an outdated status that are exposed to high rates of tree cover decline may likewise currently be threatened and thus are in need of expert-based re-evaluation.

Lower rates of recent change in threat exposure were found for cropland agriculture, urban development, and burned area. Cropland showed a median expansion of 0.95% of species' extent over 16 years and a maximum annual expansion of 2.51% of the extent for *Premna richardsiae*, native to Tanzania[35] (Red List status: Vulnerable due to deforestation and the small occupied area within the species' extent of occurrence). The built-up area showed a median expansion of 0.80% of

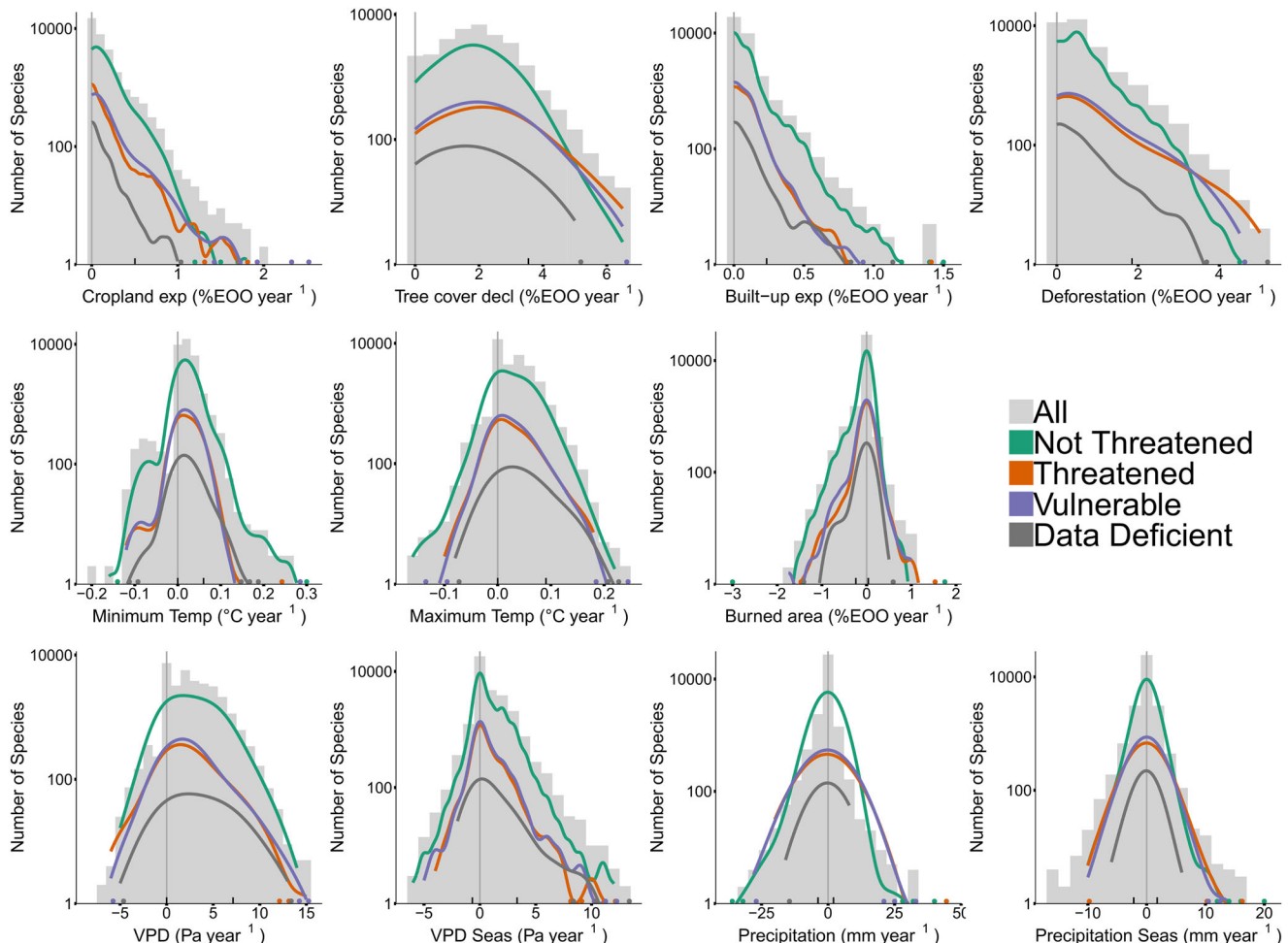

**Fig. 1 | Rates of recent change for all proxies of identified threats to trees.** Human land use changes include cropland expansion (exp), tree cover decline (decl), built-up area expansion (exp), and deforestation. Climate change includes minimum and maximum temperature (Temp), vapor pressure deficit (VPD) and its seasonality (Seas), and precipitation and its seasonality (Seas). While the bars highlight the total number of species on a log10 scale, the colored lines show the kernel density estimates per conservation status group. Rates of recent change are expressed in percentages of the species' extent per year, and for the climate variables the change is indicated by °C, Pa or mm change per year. Since this study on recent change rates has been done with data ranging from ~2000 and ~2020, one should keep in mind that the total amount of change over this time period should be multiplied by ~ 20: where a 2% decrease in extent may not seem relevant, a 40% decrease during the last 20 years can be quite alarming. The vertical gray line indicates no change. At the bottom of each plot, colored dots indicate rates of recent change per conservation status group when only one species has that value and black ticks identify the 5th and 95th quantile.

the species' extent over 20 years and a maximum annual expansion of 1.48% of the extent, for *Leucadendron strobilinum*, native to Cape Floristic Region[35] (Red List status: Near Threatened due to fire and alien invasions). Burned area had a median annual change of zero but a maximum annual increase of 1.74% of the extent for *Eucalyptus ceracea*, native to Western Australia[35] (Red List status: Least Concern), and a maximum annual decrease of 2.88% of the extent for *Pyrostria lobulata* that is native to Rwanda, Tanzania and Zambia[35] (Red List status: Least Concern).

Recent climate change has typically caused species' extents to become warmer, drier, and more seasonal in precipitation and drought, affecting species from the tropics to the subarctic over the last 20 years. Annual increases in minimum temperature were experienced by 67.6% of all included tree species, with a maximum rise of 0.31 °C for *Pinus peuce* native to the Balkan Peninsula and introduced further north in Europe[35] (Red List status: Near Threatened). Annual increases in maximum temperature were experienced by 59.8% of all included tree species, with a maximum of 0.24 °C for *Trichilia bullata*, native to the Brazilian Amazonas[35] (Red List status: Vulnerable). Annual increases in VPD were experienced by 74.6% of all included tree species, with a maximum of 15.5 Pa for *Caryodaphnopsis cogolloi*, native to

Colombia[35] (Red List status: Endangered due to wood harvesting and mining). Annual increases in VPD seasonality were experienced by 39.7%, with a maximum of 13.1 Pa for *Macrolobium urupaense*, native to Brazil[35] (Red List status: Data Deficient). Annual increases in precipitation seasonality were experienced by 23.0%, with a maximum of 20.2 mm for *Hypericum bequaertii* native to Kenya, Uganda, and Democratic Republic of Congo[35] (Red List status: Least Concern). Precipitation increased and decreased for a similar number of species (22.2% vs. 28.5%), with a maximum annual increase of 43.3 mm for *Magnolia lenticellata* (Red List status: Endangered due to logging) and a maximum yearly decrease of 37.9 mm for *Guatteria argentea* (Red List status: Least Concern), both species being native to Colombia[35].

Species' rates of change were most correlated for land use factors, where species exposed to strong cropland expansion likewise tended to increase in urban area (r = 0.22) and decrease in burned area (r = −0.29) (Supplementary Fig. 1). Further, species highly exposed to tree cover decline additionally face climate change (847 species), deforestation (235), cropland expansion (41), or burned area changes (72) (Supplementary Fig. 2). In turn, species highly exposed to changes in burned area also faced substantial exposure to climate change (998), cropland expansion (500), or deforestation (150)

(Supplementary Fig. 2). Note that changes in tree cover decline (i.e., any tree cover reduction) and deforestation (i.e., from ≥50% tree cover to 0%) were only 37% correlated, highlighting that the difference between the two threats is not only theoretical in terms of definition but also practical in terms of capturing different global dynamics.

## Conservation status groups along threat gradients

We assessed the congruence of IUCN Red List conservation status groups with the quantified changes in threats after creating a Threatened group (all species listed as Endangered and more threatened classes), a Vulnerable group (all species listed as Vulnerable), a Not Threatened group (all species listed as Near Threatened or Least Concern, note these species are not of 'no concern'), and a Data Deficient group (all species listed as Data Deficient). Across all conservation status groups, most species are exposed to low rates of recent change, and a minority are exposed to high rates of recent change (Fig. 1). We found all conservation status groups to be present along the entire exposure gradient for most threats, even at the extreme ends. As an example, there were 2623 species facing degradation in more than 50% of their extent due to tree cover decline or deforestation in the last two decades, of which 32.7% were listed as Vulnerable or Threatened, 32.4% as Not Threatened, and 34.8% as Data Deficient or Not Evaluated. Similarly, of all species that were proposed as candidates for prioritization due to climate change by our analysis ($n = 11,645$), only 9.0% were listed as Threatened, and 2.0% were listed as Data Deficient, the latter representing an alarming 43.7% of all Data Deficient species in this study.

To better understand the distribution of IUCN Red List statuses and rates of recent changes in threats, we categorized species according to their extent, an important factor in the IUCN Red List assessment process where species with a smaller extent are more likely to be listed as threatened. We found that most species with a large extent ($> 20,000$ km$^2$) were Not Threatened (60.9%), while most species with a small extent ($<5000$ km$^2$) were Threatened (25.0%) or Not Evaluated (38.6%). Nevertheless, we still found Not Threatened and Vulnerable species of all extent sizes at the extremes of exposure to all threats (Supplementary Figs. 3–9).

Additionally, we tested how rates of recent climate change may be sensitive to the selected time window. We recalculated rates of recent climate change using 10-year time windows, matching the time window for the IUCN Red List re-evaluation. For all included climatic variables, the extreme changes became more extreme for more species in the latest decade (2010–2020) compared to the prior (2000–2010) or the combined decades (2000–2020), reflecting accelerating and stronger trends in climate change (Supplementary Figs. 4–9).

## Prioritizing tree species for IUCN Red List re-evaluation

The exposure of changes in threats is defined by the continuous rates of recent change for each threat, yet we propose ~1605 species per threat as priority candidates for IUCN Red List re-evaluation based on the 95$^{th}$ percentile threshold (as literature-defined thresholds are missing) to identify highly exposed species (Supplementary Table 1). There are 17,393 unique tree species (54.2% of the species included in our study) experiencing major and increasing threats. A large number of these species are potentially being overlooked as being at risk, as 8119 of the priority candidates are listed as Near Threatened or Least Concern on the IUCN Red List (49.1% of all species in our Not Threatened conservation status group), 312 are listed as Data Deficient (57.8% of all Data Deficient species), and 5792 are Not Evaluated (58.4% of all Not Evaluated species). On the other hand, 1521 of these priority candidates are listed as Endangered or worse on the IUCN Red List (63.9% of all species in our Threatened conservation status group), and 1649 species are listed as Vulnerable (60.8% of all species in our Vulnerable conservation status group). Further, independent of the IUCN

categories, we recorded 9741 species that occupied a minimal area (see "Methods"). This may reflect the true range size of those species, making them rare species, or an artifact of missing data in this study. Nevertheless, using average rates of recent change of co-occurring species, these tree species were overall more exposed to deforestation, tree cover decline, built-up area expansion, and changes in VPD, while they were less exposed to changes in burned area, maximum temperature, precipitation, and VPD seasonality compared to locations where none of these rare or data deficient species occur (Supplementary Fig. 10).

High densities of species exposed to high rates of recent change (i.e., the priority candidates who exceed the 95$^{th}$ percentile of a threat's rate of change) were mostly found south of the Tropic of Cancer (Fig. 2a). These hotspots were located in the South American and African (sub)-tropical moist broadleaf forests and in the (sub)-tropical regions of China, Tanzania, and Malaysia. The distribution of species exposed to high rates of recent global change relative to the number of species present shows a different pattern, with the most prominent hotspots in the (sub)-tropical regions of China, the Arctic Archipelago of Canada, and northern Russia (Fig. 2b).

We observed that the hotspot locations are different for each threat, yet they are mostly located in equatorial regions (Fig. 3). Cropland expansion and climate change most heavily affected species in South America, where the former drives a hotspot of exposed species in the Cerrado and Atlantic Forest and the latter in the Amazon. Tree cover decline and deforestation mostly affected (sub)-tropical moist broadleaf forests, most apparent in the north of coastal Central Africa and in Indo-Malaysian, respectively. Built-up area expansion was primarily affecting species in Central and South China. Changes in burned area were most apparent in the extent of species in the African (sub)-tropical broadleaf moist forest and the African and Australian (sub)-tropical grassland, savanna, and shrubland biomes.

Of the priority candidates, 22.3% were highly exposed to more than one of the six threats. As a maximum, three species were found to be exposed to five out of six threats: *Zanthoxylum mezoneurispinosum*, a species native to the Ivory Coast and Liberia[35] that is listed as Vulnerable on the IUCN Red List due to urban and agricultural expansion and the small occupied area within the species' extent of occurrence, *Gluta cambodiana*, a species native to most of Mainland Southeast Asia that is Not Evaluated on the IUCN Red List, and *Apodytes geldenhuysii*, a species native to the Cape provinces of South Africa[35] but is considered as Rare on the Red List of South African Plants (redlist.sanbi.org). We see a different pattern when considering the proximity of species facing different threats. Some species may be close to a threat in terms of distance, but the threat occurs just outside the polygon we consider a species extent. To get a better estimate of the number of threats per region, we overlay the binarized, 1 km resolution hotspot maps and found that there are areas where species are heavily exposed to five or six threats within or close to their extent, mainly in Thailand, Cambodia, Vietnam, Borneo, the coast of West Africa, and small areas in Bolivia and Brazil (Supplementary Fig. 11).

## Discussion

Our systematic data-driven threat assessment quantifies recent changes within tree species' extent on a continuous scale, which can expedite conservation assessments by prioritizing species for IUCN Red List re-evaluations, ultimately helping conservation regulations and actions[34]. While the ordering of species for re-evaluations can be performed using the continuous rates of change values per threat, we highlight that 17,393 tree species require prioritization for expert-based assessment, as they have been exposed to large changes for at least one of the GTA-identified threats to trees between 2000 and 2020. Remarkably, these species comprise more than half (54.2%) of all the tree species assessed here.

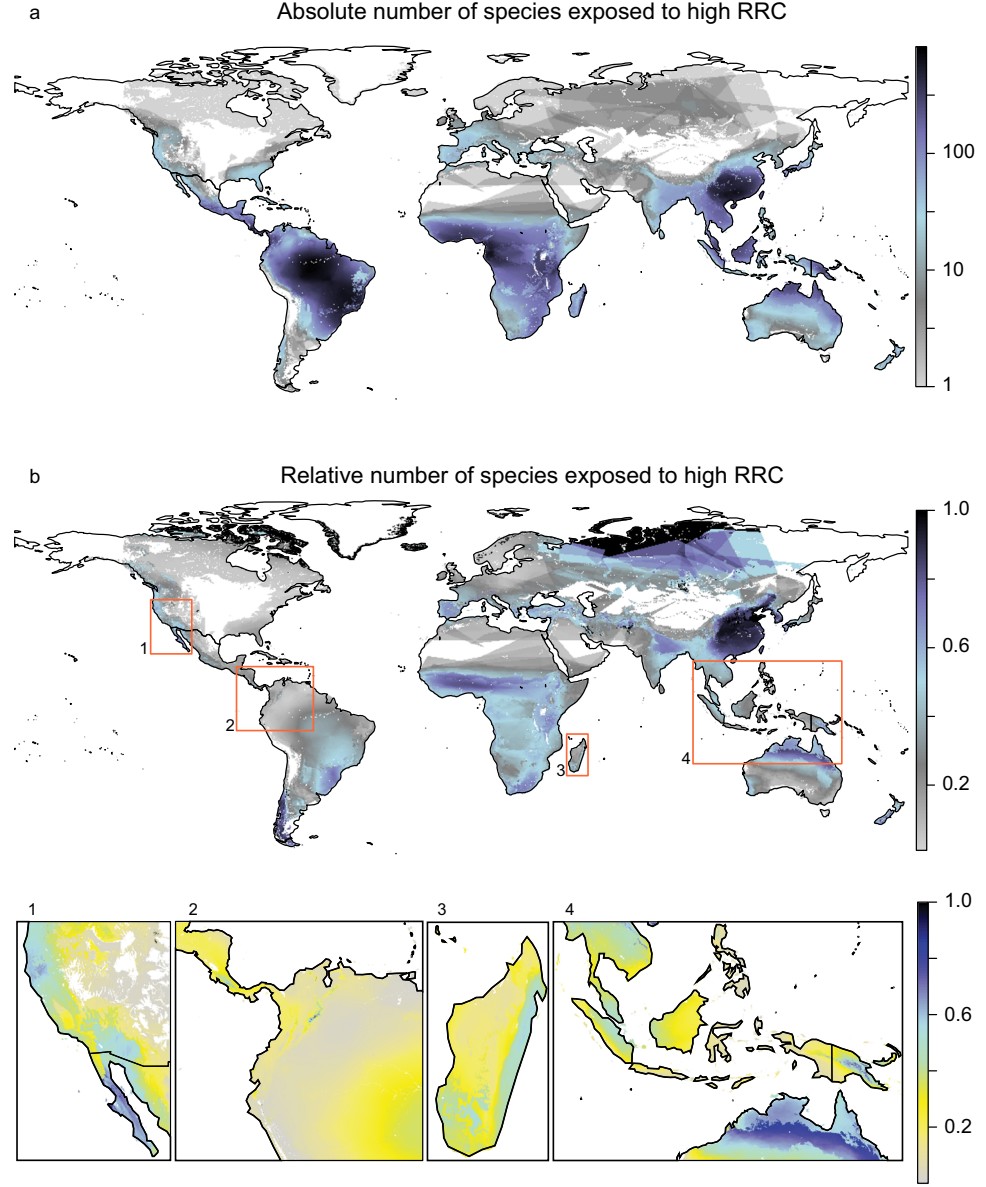

**Fig. 2 | Mapped tree species exposed to high rates of recent change (RRC).**
**a** Absolute number of tree species on a log10 scale. **b** Fraction of local number of tree species that are exposed to high RRC compare. These species have been listed as priority species for IUCN Red List re-evaluation. Colors indicate the number of species' extent overlapping per grid cell.

Our analysis shows that threats to trees may be strongly underestimated. Tree cover decline (i.e., the reduction of tree cover in any vegetation type) and deforestation (i.e., complete removal of all trees in a grid cell with originally >50% tree cover) each degraded more than 50% of species' extent for 2293 and 549 tree species, respectively, which are extreme values but not uncommon[27,33,36]. More alarming is that only ~20% of these tree species are appertained to our Threatened conservation status group. Hence, while deforestation increases the odds of being assessed as threatened for vertebrate species[37], this is not reflected by the IUCN Red List statuses for tree species[33]. This potential mismatch is not only a feature for the deforestation rate gradient, as all conservation status groups were present along the entire gradient of all threats. Additionally, of all candidate tree species that require prioritization for IUCN Red List re-evaluation, only 8.7% are listed as threatened, while these candidate species include 57.8% of all Data Deficient species and 58.4% of all Not Evaluated species, suggesting many species in these groups are in fact at risk from anthropogenic pressures. Borgelt et al.[38] predicted that 56% of all Data Deficient species over all taxa were threatened by extinction, suggesting that the underestimation of threats indicated by our results may be generalized to other organism groups.

Similar to visible land use changes, the impact of climate change on trees is likely underestimated. Overall, we found climate change to impact trees mostly through increased temperatures and drought, as expected under global warming[39]. While some species are exposed to high rates of climate change and are listed as Threatened, e.g., *Clermontia clermontioides*, a species with a small extent native to Hawaii[35] with among the top 5 highest rates of change in precipitation and precipitation seasonality, only very few species are listed as Threatened due to climate change. This contrasts with increasing evidence that climate change is causing forest dieback[40–42] and reduced forest resilience[43]. Even widespread species, like *Quercus robur*, *Picea obovata*, *Pinus sibirica*, and *Abies sibirica* are already facing major diebacks from climate change[42,44], but this has not led to elevated conservation concerns on the IUCN Red List.

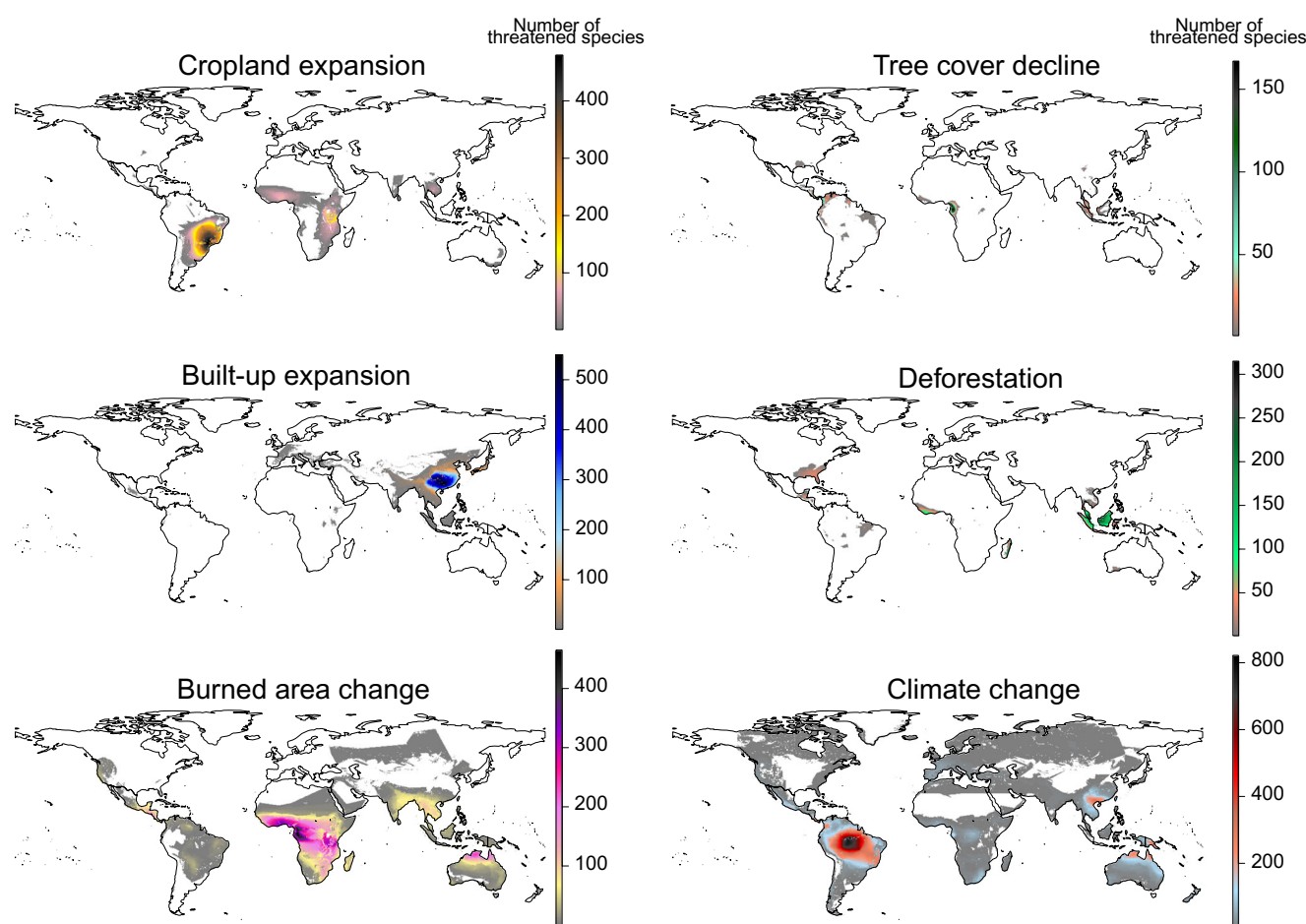

**Fig. 3 | Hotspots of tree species that are highly exposed to each threat.** Burned area change and climate change maps combine species that are facing increases and decreases. Colors indicate the number of species' extent overlapping per grid cell. A single threatened species is indicated in dark gray.

Further, species like *Ardisia cabrerae* and *Ardisia mcphersonii*, two species with a small extent that are amongst the top 10 species with the most extreme climate change in their extent according to our analysis, have been listed as Least Concern in 2020 without any mention of climate change in their IUCN Red List statuses. However, whether climate change effects are problematic depends highly on species' adaptive and dispersal abilities, generational inertia and how climate change is spatially distributed within species' extent[45,46]. Nevertheless, rising temperatures may induce heat stress (although partially offset by increases in water use efficiency[47,48]), and extreme drought can have severe and partly unknown consequences for trees in general[40,49–52], putting ecosystems across entire regions at risk, e.g., the boreal biome and tropical wet lowlands[42,53–55]. Indicating the rising pressure from climate change, we found climate change rates within tree species' extent to have intensified in 2010–2020 relative to the previous decade, in line with other studies[56,57], adding to the concern over risks linked to this threat. While effects of climate change may be difficult to estimate in terms of population trend changes or loss of locations as required to update IUCN Red List statuses, species exposed to high rates of climate change should be re-evaluated regularly in case any effects do become observable. Hence, the species prioritization based on rates of climate change can help identify species that require higher priority in the decadal IUCN Red List re-evaluations.

Threats are distributed unequally worldwide creating hotspots of highly exposed species in areas where threats are most severe[11,21,22,58]. For example, the gradual reduction in burned area over the last two decades is a global phenomenon[59] that creates shifts in inter-specific competition[60,61], increases the risk of megafires due to biomass built-

up from promoted succession[62,63], and halts reproduction or regeneration in fire-dependent species[64,65]. We find hotspots of highly exposed species in areas with extreme changes in burned area (Sub-Saharan Africa, Australia)[59]. However, we did not find a hotspot of exposed species in other areas with large changes in burned area (South America, Southern border of Eurasian boreal forests). Why changes in threat are more easily spotted around the equator can be explained by the unit for threat quantification (percentage of extent), the threshold selection (relative to other species), and the high diversity of small-ranged species at low latitude and fewer larger-ranged species at higher latitudes[66]. Species in other areas could also be affected by threats even when they do not show on these maps. Nevertheless, the identified hotspots are locations with the highest tree species diversity exposed to great change, suggesting these sites and/or species require conservation attention.

Certain threats act in synergy, indicating a need for caution when interpreting threats separately[67,68]. For example, there are two issues that need to be considered. First, some synergies are causal. Agricultural expansion and intensification likely are primarily drivers of decline in fire activity, suggesting the effectiveness of fire management in agricultural areas[59]. Such synergies may escalate effects, such as severe drought-induced die-offs under warmer conditions in areas with local tree cover decline[69,70]. Second, attributing change can be complex, particularly with issues like deforestation and tree cover decline. Tree loss can be directly human-induced, like timber harvesting, land clearing for mining, or livestock grazing. Alternatively, it can be indirectly caused by factors like disease, climatic stress, or fire due to anthropogenic climate change, globalization and/or land

use[71–74]. Hansen et al. provided insight into this complexity, linking deforestation causes to forestry practices in tropical areas and fire in boreal forests[13], where the latter is often triggered by timber harvest[75]. We report similar interconnections between deforestation, tree cover decline and changes in fire regimes, but pinpointing the actual origin or defining amplified effects remains challenging.

Using the dynamics of threats to prioritize species for IUCN Red List re-evaluations could ultimately promote timely conservation actions and policy changes[25,26]. For example, the quantification of rates of change in various threats to tree species can inform the experts performing the species-specific assessments on the need for a routine re-assessment. Additionally, when proven relevant and accurate enough, rates of recent change values might eventually also be included in the decision tree that leads up to the final species extinction risk status. The combination of high rates of recent change within tree species' extent, the likely amplification from spatial overlap in threats, and the mismatch with current species' IUCN Red List statuses implies that (1) tree species labeled as Not Threatened are not sensitive to the studied threats (yet genetic diversity may be decreasing)[76] or (2) these species require a IUCN Red List re-evaluation. On the other hand, the 17,393 species that were identified for prioritization include only 63.9% of all Threatened species and 60.8% of all Vulnerable species, suggesting that (3) threatened species with lower rates of recent change values may be more sensitive to specific threats or (4) threats can be harmful even with low rates of recent change. Future studies could look for the origins of these discrepancies, making the incorporation of changes in threat into the Red List assessment more comprehensive.

Our analyses show that ~17,000 tree species are experiencing increasing exposure to global change stressors. Due to the convergence of threats, these species will likely exhibit intensified responses as a result of synergistic effects. Consequently, these species may be at a higher risk of extinction than indicated by their current IUCN Red List statuses, implying a likely global underestimation of risks to tree species. The IUCN Red List assessments are highly valuable[25,26] yet subject to constraints of time, cost, and data accessibility[17,77,78]. These limitations can be overcome by combining expert-based knowledge with data driven approaches. The threat change quantification on a continuous scale can provide such additional information for experts, not to replace in-depth expert-based assessments but to help prioritize species for these time-consuming assessments despite scarce resources, as our method is transparent and flexible[79]. For example, the ~17,000 species exposed to recent global change can be ranked according to the number of threats species are exposed to, or the rate of change of a particular (set of) threat(s), to help identify species most in need of re-evaluation, e.g., considering additional criteria. Our method can easily be extended to other taxa and can act as an early-warning tool, especially for inconspicuous threats like climate change, allowing for a systematic approach to expedite and broaden IUCN Red List assessments in this time of global change. This combinations of recognizing rapid changes in threats within species' extent and expert-based IUCN Red List (re-) evaluations is ultimately key to continuous monitoring and conservation of biodiversity given the fast rates of global change and biodiversity loss currently being experienced.

## Methods
### Species selection
The rate of recent change quantification included all plant species that are considered to have a tree growth form conform the following definition: "a woody plant with usually a single stem growing to a height of at least two meters, or if multi-stemmed, then at least one vertical stem five centimeters in diameter at breast height"[24]. We extracted species names from the world tree species checklist

GlobalTreeSearch v.1.6 on the 10th of May 2022 (www.bgci.org[24]) and standardized them using the online Taxonomic Name Resolution Service[80].

### Species occurrences
Species' occurrence records were obtained from the TREECHANGE dataset[81]. This data originates from five open-access, publicly available data aggregators: the Global Biodiversity Information Facility[82], the public domain of the Botanical Information and Ecological Network v.3 (http://bien.nceas.ucsb.edu[83]), Latin American Seasonally Dry Tropical Forest Floristic Network (www.dryflor.info[84]), RAINBIO database (www.rainbio.cesab.org[85]), and the Atlas of Living Australia (ALA; www.ala.org.au). Serra-Diaz et al.[81] labeled all records depending on their quality, with e.g., AAA and H being the most and least reliable occurrence record, respectively. From the complete dataset, we selected occurrence records with the labels AAA, AA, A, B, C, thereby excluding records with 'Geographic coordinate issues and environment issues', 'missing environmental information or unlikely environment (botanic garden)', 'Unknown range', 'Duplicate records' or 'Missing coordinates'. We did not exclude occurrences that were identified to have only geographic coordinates issues or only environmental issues because either can be misleading when a species has a low overall number of occurrences, but together they are more likely to be erroneous. We included 41,835 tree species with 8,408,454 occurrence records covering 10,789 grid cells on a 0.1-degree resolution (Supplementary Fig. 12).

### Extent of occurrence
Rates of recent change were quantified per species within their extent of occurrence, which is used "to measure the degree to which risks from threatening factors are spread spatially across the taxon's geographical distribution" and is calculated as the smallest area (km²) encompassing all occurrence records of a species[86]. Here, we estimated species' extent by creating minimum convex polygons with 95 percent of the species occurrence records using the *mcp* function in the 'adehabitatHR' R package[87]. The minimum convex polygons were projected on the World Geodetic System (WGS84) at 0.01-degree resolution (~1 km at the Equator), the highest resolution of threat layers used in this study. The datapoint inclusion threshold was selected as an analogous approach to geographic range building, and to help protect against undue influence of apparent outliers from e.g., country center points, when experimenting with the data. While apparent outliers need not be true errors, e.g., collection efforts are very uneven and could generate such seeming outliers, for many purposes we preferred to remove points more conservatively, as we deemed it more relevant in this study to create conservative estimated species ranges rather than including geographically remote occurrences that may be beyond the (natural) range. On the other hand, we included occurrence records from the species' native and non-native range, since both could provide opportunities for species' persistence in the context of global change[88].

Species with a minimally occupied area, defined here as species with an area of occupancy (i.e., the area within a species' extent of occurrence which is occupied by that species[86]) smaller than 10 km² defined on a 2 × 2 km grid or species with less than five occurrences, were listed for IUCN Red List assessment prioritization due to their small occupied area (n = 9741). The threshold of five occurrences is selected based on requirements for the hull calculation. The threshold of 10 km² is used by the IUCN Red List to define Critically Endangered species[86]. Covering such a small area suggests that species should be prioritized for IUCN Red List re-evaluation by default because (a) they are very rare or (b) the data on this species is not complete making data-driven approaches unreliable. In addition, we set these thresholds because retrieving single grid cell data from global maps may be inaccurate due to uncertainty in the models producing these data maps.

## Specifying species' extent

The minimum convex polygon of species may include large areas where the species is not present, especially for widespread species. For example, *Erythrina velutina* occurs in the seasonally dry tropical biome[35] yet a large part of the Amazonian rainforest would be included in its extent of occurrence while it does not occur there. Therefore, area that was climatically unsuitable for each tree species specifically was masked from species' minimum convex polygon (Fig. 4). This will allow for more accurate quantification of rates of recent threat changes in species' extent, as we only consider areas where the species can potentially occur. With similar reasoning, we removed waterbodies as they are considered unsuitable habitat for trees. We refer to these updated species' extent of occurrences as species' extents in order to identify the difference. The size of species' extent was calculated by summing the proportion of each suitable grid cell covered by the species' polygon multiplied by the area of each specific grid cell, which were retrieved using the *exact_extract* function in the 'exactextractr' R package[89].

Waterbodies were identified by Modis MOD44W for the year 2015[90], which we resampled from the original 250 m resolution to the resolution of the climate zone data (~1 km, see below) using bilinear interpolation. In case of mangrove or marshland tree species, this may lead to an underestimation of their extent, but drought and/or a rising sea level threaten these species more than other tree species, making the underestimation an acceptable methodological decision.

Tree species' climatically suitable habitat was derived by considering all unique climate zones that cover at least 5% of all grid cells within a 1-km radius buffer around each species' occurrence records (only considering grid cells that cover at least 25% of the buffer area around the occurrence record). This threshold was selected to remove occurrences that occur in microclimates within other climate zones and potential inaccuracies that remained after data cleaning (e.g., old occurrence records, country centroids). We selected the Köppen Geiger climate classification raster from 1981 to 2010 (kg2) at 0.0083 degree resolution (~1 km² grid cells at the Equator) from Chelsa 2.1[14], as these are the most updated climate zone maps and the time frame best

reflects the time period where the majority of the species' occurrence records was gathered.

## Published IUCN assessments

We retrieved tree species' IUCN assessments on the 12th of January 2023 using the *iunc_summary* function in the 'taxize' R package[91]. The considered IUCN Red List statuses, from most threatened to least threatened, are: Extinct (EX), Extinct in the wild (EW), Regionally extinct (RE), Critically Endangered (CR), Endangered (EN), Vulnerable (VU), Lower risk conservation dependent (LR/cd), Near threatened (NT and LR/nt), Least concern (LC and LR/lc). Data Deficient (DD) species and species without an evaluation (Not Evaluated) are considered separately in this study.

## Threat layers

The Global Tree Assessment identified nine tree threats (Table 1)[5]. We aimed to determine the rates of recent change per threat for each species, calculated as a percentage of species' extent (km²) that became unsuitable, damaged, or underwent a change in trend over the last 20 years (between ~2000 and ~2020; Table 1). Therefore, data for each threat required global coverage and at least two time-steps (one around 2000 and one around 2020). Neither these data nor suitable proxies were available for livestock farming, energy production and mining, wood and pulp plantation, and invasive and other problematic species, which is why these four threats were not considered in this study. We included deforestation as an additional threat in this analysis, which we argued could be used as a proxy for impacts of livestock farming, energy production and mining, and tree plantations since they are typically associated with completely removing closed tree stands. We acknowledge that deforestation, the measure of forest loss to be distinguished from forest degradation or logging, will find (partial) overlap with other land use change threats, i.e., crop agriculture and urban development. To include climate change, we quantified drought and temperature shifts using yearly minimum and maximum temperatures, mean annual vapor pressure deficit (VPD),

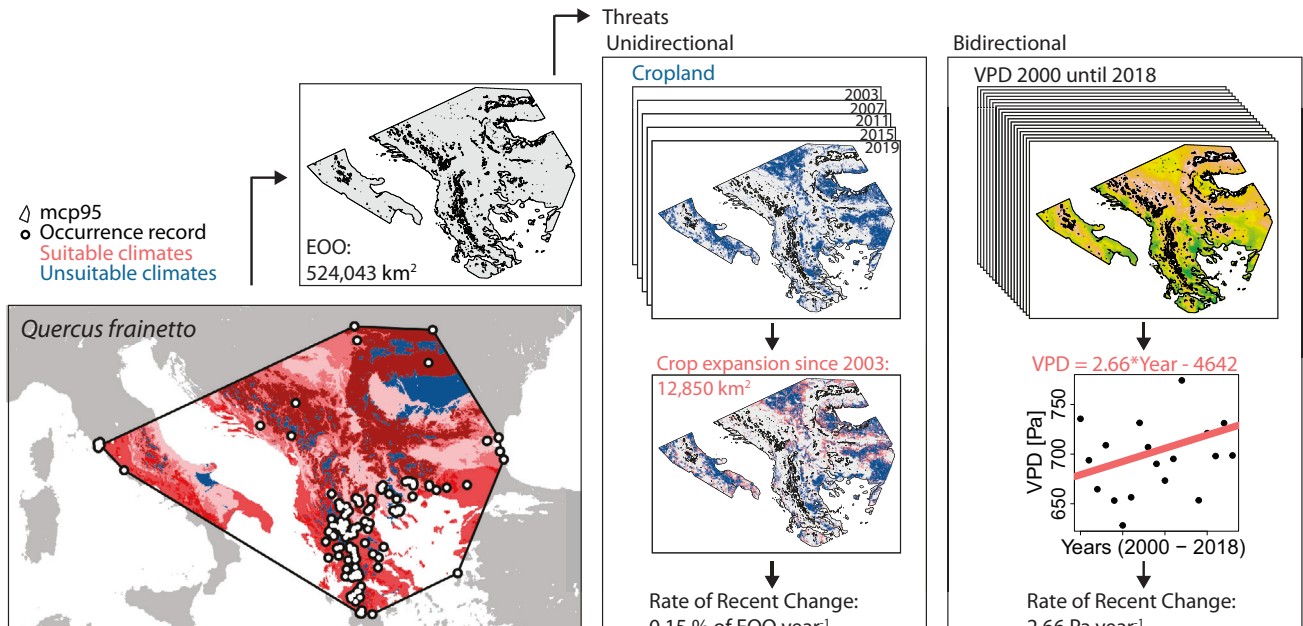

**Fig. 4 | Stepwise overview of threat analysis.** The tree species presented, as an example, is *Quercus frainetto*. The species' extent is defined as land area and suitable climate, in which threats are measured. Cropland expansion is used as an example to show the process for unidirectional threats to reach a rate of recent change value. Vapor pressure deficit (VPD) is used as an example to show the

process for bidirectional threats to reach a rate of recent change value. mcp95 stands for minimum convex polygon using 95% of the occurrence records. Please note that we changed the resolution of the crop expansion layer for illustrative purposes.

**Table 1 | Overview of GTA-identified threats to trees[5] and their proxies used in this study**

| | Threat (% of affected tree species)[5] | Proxy | Data info | RRC [% of extent year[-1]] calculation |
|---|---|---|---|---|
| Unidirectional change | Crop agriculture (29%) | Cropland expansion<br>The cropland classification was given to grid cells with land used for annual and perennial herbaceous crops for human consumption, forage (including hay) and biofuel, excluding perennial woody crops, permanent pastures and shifting cultivation. Land abandonment was not considered.<br>Cropland extent gain was calculated as the sum of areas of grid cells defined as no cropland or no data that, at any time comparison, changed to an agriculture defined grid cell. Note that area already being cropland in 2000-2003 was not considered in this analysis, hence we consider 2003 as the baseline year. | GEE[11,58]:<br>users/potapovpeter/Global_cropland<br>Time range: 2003–2019<br>Original resolution: 30 meters<br>Number of layers:5<br>Original data values:<br>0 – no cropland or no data<br>1 – cropland | $map_{post2003} = map_{2004\text{-}2007} + map_{2008\text{-}2011} + map_{2012\text{-}2015} + map_{2016\text{-}2019}$<br>Binarize $map_{post2003}$ so that cells converted at any timestep post 2003 have a value of 1<br>$map_{expansion} = map_{post2003} - map_{2000\text{-}2003}$<br>$Rate = \dfrac{\sum cell\ area_{value==1}}{EOO_{area}} /16$ |
| | Overexploitation (27%) | Tree cover decline<br>Tree cover (TC) is the percentage of horizontal ground in each 30-m pixel covered by woody vegetation greater than 5 meters in height.TC decline was calculated as the sum of areas of grid cells that have seen a decrease (minimum 5%) in TC at any time comparison. Stable or increases in tree cover were not considered. Although using this variable to quantify overexploitation or selective logging has been advised against in the past[97,98], data has improved and the use of tree cover decline will tell a more complete story to forest disturbance and degradation[99–101]. | GEE[12]: NASA/MEASURES/GFCC/TC/v3<br>Time range: 2000–2015<br>Original resolution: 30 meters<br>Number of layers: 4<br>Original data values: TC (%) | $TC_{2000} - TC_{2005}$<br>$TC_{2005} - TC_{2010}$<br>$TC_{2010} - TC_{2015}$<br>Combine these three difference maps using maximum, resulting in each cell that underwent a reduction in TC at any timestep having a negative value.<br>$Rate = \dfrac{\sum cell\ area_{value>1}}{EOO_{area}} /15$ |
| | Urban development (13%) | Built-up area expansion<br>The built-up classification was given to grid cells with any man-made land surface associated with infrastructure, commercial and residential land uses.<br>Built-up area expansion was calculated as the sum of grid cell area defined as no built-up land in the year 2000 and changed to a built-up defined grid cell in the year 2020. Note that area already being built-up in 2000 was not considered in this analysis. | GEE[11]:<br>projects/glad/GLCLU2020/Builtup_type<br>Time range: 2000–2020<br>Original resolution: 30 metersNumber of layers: 1<br>Original data values:<br>0 – no built-up area<br>1 – stable built-up area<br>2 – built-up expansion | $Rate = \dfrac{\sum cell\ area_{value==2}}{EOO_{area}} /20$ |
| | Habitat loss | Deforestation<br>Deforestation was defined as the complete removal of the tree canopy at the Landsat pixel scale between 2000 and 2020. The forest classification was given to grid cells with ≥ 50% of the grid cell area filled with vegetation ≥ 5 m and refers to tree cover and not land use. Deforestation was calculated as the sum of areas of grid cells defined as forest loss between the year 2000 and 2020. | GEE[13]:<br>UMD/hansen/global_forest_change_2021_v1_9<br>Time range: 2000 – 2020<br>Original resolution: 30 meters<br>Number of layers: 1<br>Original data values:<br>0 – no forest or no forest loss<br>1 – forest loss | $Rate = \dfrac{\sum cell\ area_{value>1}}{EOO_{area}} /20$ |
| Bidirectional change | Fire and fire suppression (13%) | Burned area<br>Using surface reflectance in the Near Infrared band from MODIS, burned pixels are indicated in addition to the estimate date of burn.A change in fire regime was calculated as a significant trend over time in the sum of grid cell area defined as burned in each year. Though other aspects of fire regimes are important for tree mortality, we wanted to keep the amount of detail for each threat similar and selected a variable used by other global studies[36]. In addition, the relatively short time period is able to capture trends in burned area. | GEE[15]: ESA/CCI/FireCCI/5_1<br>Time range: 2001 – 2020<br>Original resolution: 250 meters<br>Number of layers: 20<br>Original data values:<br>burn date (day of the year) | Per year, burned area within extent: $\sum cell\ area_{value>0}$. Fit a linear model of area burned over the years. If the slope is significant, the slope indicates the RRC. If the slope is not significant, the RRC is set to 0. |

**Table 1 (continued) | Overview of GTA-identified threats to trees5 and their proxies used in this study**

| Threat (% of affected tree species)[5] | Proxy | Data info | RRC [% of extent year⁻¹] calculation |
|---|---|---|---|
| | burned area over time[59]. Please not that this variable is dissimilar to fire return interval. | | $\sum \frac{\text{climate variable}}{EOO, \text{number of grid cells}}$ |
| Climate change (4%) | Temperature and drought Trees may not be directly affected by small differences in temperature and drought (Vapor Pressure Deficit (VPD) and precipitation), but prolonged periods or extremer extremes can impact the individual and if not the forest around the individual in turn affecting that individual. We considered minimum and maximum temperature, VPD, VPD seasonality, precipitation and precipitation seasonality. A change in climate variable was calculated as a significant trend in yearly averages of extent over the years. Temperature values were transformed from K*10 to °C and the precipitation unit equals mm. | Source[14]: Brun et al. (2022) Time range: Temperature 2000 – 2019 VPD 2000 – 2018 Precipitation 2000 – 2019 Original resolution: 4638.3 meters Number of layers: 6 variables * 20 years Original data values: Temperature (K*10) VPD (Pa) Precipitation accumulation (kg.m⁻²) | Per year, climate variable within extent: Fit a median-based linear model of yearly averaged climate variables over the years. If the slope is significant, the slope indicates the RRC. If the slope is not significant, the RRC is set to 0. Note, the unit for this threat is not [% of extent year⁻¹] but [°C year⁻¹] [mm year⁻¹] or [PA year⁻¹] |

The rates of recent change (RRC) indicate the percentage of the tree species' extent (surface area of species' minimum convex polygon minus the surface area of surface water in that polygon minus the area of unsuitable climate zones) that has been converted due to a threat during the indicated time range. Threats are identified as unidirectional (only increasing in threat and reversal is not considered) or bidirectional (increasing and decreasing values are considered as threat). Please note that also the area of surface water was removed from the calculated threat extent. GEE stands for Google Earth Engine and identifies the data's Earth Engine asset ID.

VPD seasonality (sd), mean annual precipitation (MAP), and MAP seasonality (sd). These data were downloaded from the CHELSA-BIOCLIM + dataset as monthly layers of temperature, VPD, and precipitation for the available years between 2000 until 2020[14]. We preprocessed these monthly layers to yearly maps using GDAL[92]. For temperature, the annual minimum and maximum were the minimum and maximum value over the 12 months. For VPD and precipitation, the mean annual maps were calculated as the average value over 12 months and seasonality was calculated as the standard deviation over the 12 months.

We want to note that interpretation of single threat layers should be done with care. When, for example, deforestation is caused by cropland expansion, the conversion of the same area is counted twice. At the same time, cropland can also expand in areas that were not previously covered by trees, in which case additional area is converted causing increased threat. This is also suggested by Potapov et al.[58], who found that only in tropical regions (Africa and South-East Asia and, to a lesser extent, South America and South-West Asia) cropland expansion followed conversion of natural vegetation (possibly deforestation), while cropland expansion replaced pastures or abandoned agricultural lands in temperate areas.

### Recent change calculations

To quantify how global change affects species' habitats, we calculated rates of recent change per threat layer within each species' extent that has been updated with suitable climate zones and excludes surface water areas. We prepared threat change maps in Google Earth Engine[93] (GEE) as described in Table 1, identifying grid cells that became unsuitable, damaged, or underwent a change in trend over the last 20 years (between ~2000 and ~2020). These binarized maps were then resampled using the 'nearest neighbor' method to the resolution of the climate zone data (~1 km²) and downloaded using the 'rgee' R-package[94] (Fig. 4). In case of threats expressed in km², we used the minimum convex polygon per species, removed waterbodies and unsuitable climate zones for the species, and extracted the area of the covered grid cells classified with 'changed'. Similar to deriving the species' extent, we took into account the proportion of grid cell area covered by the species' polygon as well as the area of each grid cell. In case of the continues climate variables, we determined the average value experienced in the species' extent. Here, the grid cell values of all suitable grid cells within the polygon were averaged using the *exact_extract* function in the 'exactextractr' R package[89] with weighted means, where the weights represent the fraction of the grid cell that is covered by the species polygon multiplied by the grid cell area.

Rates of recent change values were calculated differently for threats that we considered only to increase (unidirectional change), and that could increase and decrease (bidirectional change). The unidirectional change was assumed for all land-use change threats: crop agriculture, overexploitation, urban development, and habitat loss. This means that once a grid cell changes into an unsuitable area for trees (e.g., from non-urban to urban), the area of these cells is registered as 'changed'. Although these changes may be reversed over time, we do not consider this reversal as trees have already been cut. The land may be more subjected to human decision-making, even though this assumption excludes newly designated restoration or conservation sites and land abandonment. Unidirectional rates of recent change were then calculated as the changed area divided by the size of the extent, multiplied by 100, and divided by the number of years the threat covers, resulting in a percentage area change per year. Bidirectional change was used for fire and climate change threat layers, where we used the mean value (for climate) or sum of affected area (for fire) within the species' extent per time layer, and use significant slopes from median-based linear models based on Siegel repeated medians as a measure for rates of recent change (*mblm* function in the 'mblm' R-package[95]). Here, fire threat was still expressed in relative area change per year by dividing the slope value by the size of the extent multiplied by 100, while climate change was expressed in native units (e.g., °C) per year.

**Linking rates of recent change data and IUCN Red List statuses**

To understand how species' conservation status relates to recent changes in threats to tree species, we combined rates of recent change data with retrieved IUCN Red List statuses via species names. First, we removed 9741 species from the total of 41,835 species because they had a minimally occupied area (see above) and an additional 3 species because their minimum convex polygon was a line which does not cover any area (*Coffea kihansiensis and Ravenea moorei* listed as Critically Endangered and *Frangula inconspicua*, listed as Endangered on the IUCN Red List). Second, one additional species was removed from this analysis because the rates of recent change value for one of the threat layers was an outlier (more than 3 times the interquartile range from 1% and 99%). This species is *Quararibea pumila*, which is listed as Endangered on the IUCN Red List. We included conservation status of all remaining 32,090 species, yet we highlight that 34.6% of these species received this status before the year 2000. We lumped species' IUCN Red List statuses into three conservation status groups to (1) reduce potential inconsistencies due to data accuracy from rates of recent change, (2) reduce data and knowledge dependencies among species, and (3) facilitate interpretation of (in)consistencies between rates of recent change values and conservation statuses. These three groups are: (1) Not threatened, comprising of LC, NT, LR/lc, LR/cd, and LR/nt, (2) Vulnerable, comprising of VU, and (3) Threatened, comprising of EX, EW, RE, CR, and EN. Furthermore, we included the 'Data Deficient' status for species identified as data deficient, and unassessed species received the 'Not Evaluated' status.

For the 9741 species that were removed from the analysis because of their minimally occupied area, we estimated rates of recent changes values per threat based on the grid cells they occur in, at a 1-degree resolution. First, we calculated the mean rate of recent change value over all the species with rates of change values occurring in each of those grid cells. Second, when a species without rates of change values was present in more than one grid cell, we considered the median of all grid cell values it occurs at.

To determine how species' extent and the selected time window influenced the calculated rates of recent change, we separated the data as follows. We split the data based on species' extent using the thresholds from IUCN Red List criterion B: species with a wide extent ($>20,000 \text{ km}^2$), species with a regional extent ($5000 \text{ km}^2$ <extent $<20,000 \text{ km}^2$), and species with a local extent ($<5000 \text{ km}^2$). Additionally, we adjusted the time window to 10 years and recalculated rates of recent change for all climate variables (excluding other threats due to data limitation) between ~2000 and ~2010 and between ~2010 and ~2020. This 10-year window was chosen because the IUCN Red List needs to be re-evaluated at least every 10 years[31]. As a reporting standard in this study, we used species of all extent sizes and selected the 20-year time window (~2000 to ~2020), which allowed a general overview of recent changes in threats.

**Prioritization candidates**

We identified species per threat as priority candidates for IUCN Red List re-evaluation based on the 95th percentile threshold (as literature-defined thresholds are missing) to identify highly exposed species. Next, we overlayed the species' extents that have been updated with preferred climate zones and exclude surface water areas and rasterized them to a 0.01-degree resolution, in order to count the number of species in each grid cell and map hotspots of highly exposed species per threat.

The threshold selection to define high exposure to threat requires careful interpretation of our results, as not all threats are equally risky to trees possibly making the 95th percentile as a threshold too narrow. Additionally, this threshold does not account for interactions with physiological traits, geographic range, habitat type, or specific species' sensitivities towards each threat. More research on the refinement of the threats and thresholds may benefit all work concerning nature conservation and restoration. Still, thresholds may be impossible to define in general terms, covering all species, for such applications[96].

**Reporting summary**

Further information on research design is available in the Nature Portfolio Reporting Summary linked to this article.

## Data availability

Tree occurrence data are openly available GBIF[82], BIEN v.3[83], dryflor[84], RAINBIO[85], and ALA. The TREECHANGE[81] workflow was used for data quality assessment and control. The final occurrences used in this study are available under on figshare: 10.6084/m9.figshare.24168297. IUCN Red List statuses can be retrieved from the IUCN website (www.iucnredlist.org). Threat layers are available as presented in Table 1 of the main text. Species' 95% minimum convex polygon shape files, a dataset with rates of recent change, and lists of candidate species for prioritization for all included threats are available under CC BY 4.0 on GitHub (https://github.com/ColineBoonman/RatesOfRecentChange) and mirrored on figshare: https://doi.org/10.6084/m9.figshare.24168297.

## Code availability

We include all the R code need for this study on GitHub (https://github.com/ColineBoonman/RatesOfRecentChange) and mirrored on figshare: https://doi.org/10.6084/m9.figshare.24168297.

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

## Acknowledgements

We thank VILLUM FONDEN for economic support via JCS' VILLUM Investigator project Biodiversity Dynamics in a Changing World (grant 16549). We also consider this work a contribution to Center for Ecological Dynamics in a Novel Biosphere (ECONOVO), funded by Danish National Research Foundation (grant DNRF173 to J.C.S.). J.M.S.D. and C.M. acknowledge the support of NASA Grant 80NSSC22K0883 (J.M.S.D. and C.M.), N.S.F. 2225078 (C.M.) and ANR-21-CE32-0003 (J.M.S.D.). B.J.E., C.M. were supported by NSF awards 2225078 and 2225076. YM is supported by the Jackson Foundation and the Leverhulme Trust.

## Author contributions

C.C.F.B. and J.C.S. conceived the study. J.M.S.D., W.Y.G., B.J.E., B.M., and C.M. combined and cleaned the tree observation records. S.H. retrieved and processed global data and calculated all rates of recent change. C.C.F.B. ran the analyses and interpreted the results, with input from J.C.S., S.H., R.B. The manuscript was written by C.C.F.B., with suggestions provided by J.C.S., R.B., S.H., J.M.S.D., W.Y.G., B.J.E., B.M., Y.M., and C.M.

## Competing interests

The authors declare no competing interests.
