## [Peer Review File · Nature Communications]

More than 17,000 tree species are at risk from rapid global changeREVIEWER COMMENTS

Reviewer #1 (Remarks to the Author):

GENERAL FEEDBACK AND MAJOR COMMENTS:

The authors present a data-driven approach to assess changes in six threat factors to more than 30,000 tree species (i.e. about half of the tree species on Earth). They found a large discrepancy between their results and the number of tree species assessed as threatened by the IUCN Red List. The scope and findings make it a paper of global relevance, but I have two major concerns that would require revision (though they are somewhat related, so I could also say one major concern), which I explain in detail below. I apologize for any misunderstandings that may have occurred.

While their main finding (i.e. many more tree species are at risk compared to current IUCN statuses) may be valid, as IUCN Red List assessments do indeed often overlook some of the threat factors studied by the authors (e.g., climate change), the applied methodology does not seem robust enough to make these statements with sufficient confidence, and I do have some doubts as well about the claim that their results can be used to prioritize species for IUCN Red List reassessment. At a species-specific level, the methodology does not seem adequate to give reliable estimates of vulnerability towards the studied threats or extinction risks.

My two main concerns are related to the use of the 'extent of occurrence (EOO)' as the basis for the analysis.

My first concern is that the authors did not attempt to estimate the 'area of occupancy' (AOO), but only the 'extent of occurrence (EOO)'. In fact, they even seem to mix up both terms. At lines 418-420, the authors write "Rates of recent change were quantified per species within their extent of occurrence. The extent (sic; 'this extent' may be a better wording) measures the occupied area by the taxon and is calculated as the smallest area (km²) encompassing all occurrence records of a species."

This definition seems quite misleading, as the extent of occurrence (EOO) is different from the area of occupancy (AOO). The IUCN Red List Guidelines (https://nc.iucnredlist.org/redlist/content/attachment_files/RedListGuidelines.pdf), i.e. the same document cited by the authors after the abovementioned definition, explicitly say that the EOO should not be used as a measure of occupied area: "EOO is not intended to be an estimate of the amount of occupied or potential habitat, or a general measure of the taxon's range."

Rather, the IUCN definitions of the concept EOO and AOO (<https://portals.iucn.org/library/sites/library/files/documents/RL-2001-001-2nd.pdf>) are:

"Extent of occurrence is defined as the area contained within the shortest continuous imaginary boundary which can be drawn to encompass all the known, inferred or projected sites of present occurrence of a taxon, excluding cases of vagrancy".

"Area of occupancy is defined as the area within its 'extent of occurrence (see point 9 above) which is occupied by a taxon, excluding cases of vagrancy."

To further illustrate that the authors seem to be mixing up these two concepts, they refer to the EOO various times as 'range'. In addition, they do mention the concept 'area of occupancy' twice, but without defining it, seemingly using it as a synonym of EOO, although it was not entirely clear to me if that was what they really meant.

While this may seem just a matter of definitions, the use of EOO as the basis for the analysis is the main reason why I have doubts about the usefulness of the results, especially at a species-specific level. Let's consider the example of a species such as *Vachellia nilotica*, a dryland species that occurs from Egypt to Senegal to South Africa. While I didn't have access to the convex hulls (the link provided by the authors: <https://github.com/ColineBoonman/RatesOfRecentChange> is not working for me), this species is likely to have a convex hull that covers enormous areas where it does not occur (Sahara desert, Congo basin rainforest, etc.). Another (extreme) example in which

the convex hull (extent of occurrence) would be many times larger than the area of occupancy would be the dry forest specialist *Erythrina velutina*, which occurs in both the dry forests of NE-Brazil and those of Colombia, Ecuador and Peru, but is absent from the Amazon rainforest in between. It seems clear that any estimates of threat exposure that start from a convex hull around presence points do not seem particularly useful for assessing the vulnerability of these species towards the studied threats, as these hulls contain more area not occupied by the species than area occupied.

However, as the other parts of the methodology seem robust, the findings of the study could be made much more reliable if the authors would estimate the area of occupancy rather than the extent of occurrence, or at least attempt to get as close as possible to such an estimate. The most obvious solution would be to use species distribution models (see for example <https://esajournals.onlinelibrary.wiley.com/doi/10.1002/eap.2228>), but this may be challenging given the enormous number of species. It would also reduce the number of species that can be included, because many species will not have enough presence coordinates for them to be modelled. However any estimates of ranges of suitable habitat for species with for example only 5 presence coordinates seem highly unreliable anyway, independent of the method used, so excluding such species from the analysis could be more appropriate, rather than aiming to assess as much as possible species as possible. On the other hand such species may also be the species most in need of a IUCN Red List (re-)assessment, so I understand why the authors have chosen to include as much as possible species. An alternative (although rough) approach to estimate area of occupancy could be to intersect the species' EOO with the terrestrial ecoregions, and to exclude those ecoregions where the species is not recorded, and I am sure there are other possible solutions.

My second concern is about the use of occurrence records outside the native range of species. The authors do not seem to mention if such records were included in the convex hull or not, so presumably they used records from around the world for all species? Taking again the example of *Vachellia nilotica*, which has become common in India and Australia (often invasive), and occurs in the Americas too. There are arguments pro and contra including records outside the native range in threat assessments, but if the authors also used records outside the native range, this seems to make the use of the convex hull as basis for the analysis even more problematic (the convex hull of *Vachellia nilotica* would then cover almost the entire tropics worldwide?).

IN-TEXT COMMENTS

Line 256: 'area of occupancy': this term would need to be defined, or does it refer to the same as extent of occurrence in this case (see major comments above)?

Line 281: 'area of occurrence': confusing term as it is a mix between 'extent of occurrence' and 'area of occupancy'

Line 437: 'area of occupancy': see above

Line 439: 'small occurrence': confusing term

Fig. 3: Could the authors explain why there are no hotspots of built-up expansion in South America and especially Africa (with large expansion of urban areas underway in West-Africa, for example Lagos). Same for deforestation. Not suggesting there has been an error, but this seems counter-intuitive and I would like to understand what would be the reason behind this?

Line 299 (and elsewhere): 'range': the extent of occurrence should not be used interchangeably with the concept 'range', see major comments above.

Line 342 (and elsewhere): 'burned areas': I think this should be 'burnt'

Reviewer #2 (Remarks to the Author):

It is great to see a methodology for flagging current or recent threats to tree species that could be considered when reassessing tree species and could be overlooked.

However, I do not think that the methodology is linked specifically to the IUCN Red List criteria meaning that it is not directly applicable. Criterion B for example requires not only a small extent of occurrence or area of occupancy but also continuing declines and a small number of locations in respect to threat. Criterion A requires declines to be calculated over 3 generations, for trees normally more than 20 years.

My main concern is with conclusions. The paper calls for 16,000 "expert-based assessments", however, the vast majority of these species have been assessed in the last 10 years with input from experts through the Global Tree Assessments. The majority of the non-evaluated tree species have draft assessments completed or reviewed by experts awaiting publication. If there is no direct link between the results of this analysis and the IUCN Categories and Criteria then this is asking for work to be redone that has already recently been completed by experts. There will of course be a selection of species that have erroneous assessments.

The mention of specific species and their assessments in the text is confusing when the results of this study would not impact the results.

Least Concern is not no concern, and there are few species that are not seeing global change. For example, the two *Ardisia* species that are flagged in the conclusions, these both have recent IUCN Red List assessments published by the Colombian Plant Specialist Group. Climate change is not flagged in the assessment, but it is unclear what the effects will be to the species. This is part of the reason that climate change is so rarely used in assessments as you note in line 317, as it is hard to estimate the declines for use of criterion A.

There are many reasons that species can be Data Deficient, including taxonomic. Data Deficient species are those for which an LC and a CR assessment could be equally plausible.

It would be interesting to try this analysis on the maps published on the IUCN Red List as these are cleaned maps.

Comments on specific lines:

Line 118 - I do not think this is true. IUCN Red List reevaluations are not prioritised based on likelihood of a different outcome. If there has been a negative change or errors were made, reassessments can happen faster.

Line 448 - it is not possible to use assessments conducted under synonymous names unless homotypic, not clear from the text if this included heterotypic synonyms

Line 538 - LR/cd - should be considered NT and not Threatened (this category no longer exists).

Reviewer #1 (Remarks to the Author)

GENERAL FEEDBACK AND MAJOR COMMENT 1:

The authors present a data-driven approach to assess changes in six threat factors to more than 30,000 tree species (i.e. about half of the tree species on Earth). They found a large discrepancy between their results and the number of tree species assessed as threatened by the IUCN Red List. The scope and findings make it a paper of global relevance, but I have two major concerns that would require revision (though they are somewhat related, so I could also say one major concern), which I explain in detail below. I apologize for any misunderstandings that may have occurred.

While their main finding (i.e. many more tree species are at risk compared to current IUCN statuses) may be valid, as IUCN Red List assessments do indeed often overlook some of the threat factors studied by the authors (e.g., climate change), the applied methodology does not seem robust enough to make these statements with sufficient confidence, and I do have some doubts as well about the claim that their results can be used to prioritize species for IUCN Red List reassessment. At a species-specific level, the methodology does not seem adequate to give reliable estimates of vulnerability towards the studied threats or extinction risks.

My two main concerns are related to the use of the 'extent of occurrence (EOO)' as the basis for the analysis. **My first concern is that the authors did not attempt to estimate the 'area of occupancy' (AOO), but only the 'extent of occurrence (EOO)'. In fact, they even seem to mix up both terms.** At lines 418-420, the authors write "Rates of recent change were quantified per species within their extent of occurrence. The extent (sic; 'this extent' may be a better wording) measures the occupied area by the taxon and is calculated as the smallest area (km²) encompassing all occurrence records of a species."

This definition seems quite misleading, as the extent of occurrence (EOO) is different from the area of occupancy (AOO). The IUCN Red List Guidelines (https://nc.iucnredlist.org/redlist/content/attachment_files/RedListGuidelines.pdf), i.e. the same document cited by the authors after the abovementioned definition, explicitly say that the EOO should not be used as a measure of occupied area: "EOO is not intended to be an estimate of the amount of occupied or potential habitat, or a general measure of the taxon's range."

Rather, the IUCN definitions of the concept EOO and AOO (<https://portals.iucn.org/library/sites/library/files/documents/RL-2001-001-2nd.pdf>) are: "Extent of occurrence is defined as the area contained within the shortest continuous imaginary boundary which can be drawn to encompass all the known, inferred or projected sites of present occurrence of a taxon, excluding cases of vagrancy". "Area of occupancy is defined as the area within its 'extent of occurrence (see point 9 above) which is occupied by a taxon, excluding cases of vagrancy."

To further illustrate that the authors seem to be mixing up these two concepts, they refer to the EOO various times as 'range'. In addition, they do mention the concept 'area of occupancy' twice, but without defining it, seemingly using it as a synonym of EOO, although it was not entirely clear to me if that was what they really meant.

Apologies for this lack of clarity in our writing. We indeed have a strong focus on range concepts. We believe we have a good understanding of both EOO and AOO, and accordingly we edited the text to keep the terminology clean. We now define clearly the terms EOO and AOO, and do not use the terms species habitat and species range. In short, we use the extent of occurrence as a basis of our work, whereas the area of occupancy is only used to determine highly restricted species which in turn are treated differently in this study (see manuscript and answer to Major Comment 3). Few examples are:

"... species with a small extent (<5,000 km²) ..." (lines 216)

"..., species like *Ardisia cabrerai* and *Ardisia mcphersonii*, two species with a small extent ..." (lines 309)

“Rates of recent change were quantified per species within their extent of occurrence, which is used “to measure the degree to which risks from threatening factors are spread spatially across the taxon’s geographical distribution” and is calculated as the smallest area (km²) encompassing all occurrence records of a species.” (lines 413 – 416)

“Species with a minimally occupied area, defined here as species with an area of occupancy (i.e., the area within a species’ extent of occurrence which is occupied by that species⁸⁶) smaller than 10 km² defined on a 2 x 2 km grid or species with less than five occurrences, were listed for IUCN Red List assessment prioritization due to their small occupied area (n=9,741).” (lines 429 – 432)

MAJOR COMMENT 2:

While this may seem just a matter of definitions, **the use of EOO as the basis for the analysis is the main reason why I have doubts about the usefulness of the results, especially at a species-specific level.** Let’s consider the example of a species such as *Vachellia nilotica*, a dryland species that occurs from Egypt to Senegal to South Africa. While I didn’t have access to the convex hulls (the link provided by the authors: <https://github.com/ColineBoonman/RatesOfRecentChange> is not working for me), this species is likely to have a convex hull that covers enormous areas where it does not occur (Sahara desert, Congo basin rainforest, etc.). Another (extreme) example in which the convex hull (extent of occurrence) would be many times larger than the area of occupancy would be the dry forest specialist *Erythrina velutina*, which occurs in both the dry forests of NE-Brazil and those of Colombia, Ecuador and Peru, but is absent from the Amazon rainforest in between. It seems clear that any estimates of threat exposure that start from a convex hull around presence points do not seem particularly useful for assessing the vulnerability of these species towards the studied threats, as these hulls contain more area not occupied by the species than area occupied.

However, as the other parts of the methodology seem robust, the findings of the study could be made much more reliable if the authors would estimate the area of occupancy rather than the extent of occurrence, or at least attempt to get as close as possible to such an estimate. The most obvious solution would be to use species distribution models (see for example <https://esajournals.onlinelibrary.wiley.com/doi/10.1002/eap.2228>), but this may be challenging given the enormous number of species. It would also reduce the number of species that can be included, because many species will not have enough presence coordinates for them to be modelled.

The reviewer is correct that these two examples present issues, that may occur for many other species. We agree that large areas within species’ EOO are not occupied and will never be part of the species’ realized niche. Therefore, **we changed the methods by removing unsuitable climate** (i.e. climate zones that are not occupied by the species) from species’ EOO. According to the changes in method, we adjusted the figures and results. In addition, since the area we now consider as a species ‘range’ does not have a clear term, we **changed all wording** from ‘extent of occurrence’ to ‘extent’. We agree that the term EOO does not fit our methods, yet it is also far from AOO, since vast areas are still included without knowing if the species really occupies those areas. Our approach resembles the methodology of AOH (area of habitat), but since we use climate instead of habitat we also do not feel comfortable using that term. We highlight this in the text, too: “We refer to these updated species’ extent of occurrences as species’ extents in order to identify the difference.” (lines 447 – 448)

Below, you can find the new extent maps for the two example species and a detailed description of our reasoning for the change in methods and why we opted for the specific adjustments we made guided by some quotes from the IUCN Red List guidelines, referred to as ‘the guidelines’: Guidelines for Using the IUCN Red List Categories and Criteria. Version 15.1. IUCN Standards and Petitions Committee. https://nc.iucnredlist.org/redlist/content/attachment_files/RedListGuidelines.pdf.

Figure *Vachellia nilotica*. Top: Suitable climate zones (pink shades) and unsuitable climate zones (blue) within the species' 95% minimum convex polygon (black line). Dots represent the occurrence records of this species. In the previous version of the manuscript, the EOO would represent the entire colored areas within the polygon. In the new version of the manuscript, the EOO is represented by the pink-shaded areas only. Bottom: EOO, highlighted in grey, as following the new methodology described here and in the new version of the manuscript.

Figure *Erythrina velutina*. Left: Suitable climate zones (pink shades) and unsuitable climate zones (blue) within the species' 95% minimum convex polygon (black line). Dots represent the occurrence records of this species. In the previous version of the manuscript, the EOO would represent the entire colored areas within the polygon. In the new version of the manuscript, the EOO is represented by the pink-shaded areas only. Right: EOO, highlighted in grey, as following the new methodology described here and in the new version of the manuscript.

We considered the species' EOO as the basis of our entire analysis, because "The intent behind this parameter is to measure the degree to which risks from threatening factors are spread spatially across the taxon's geographical distribution" (page 49 of the guidelines). We see now that this is not an ideal metric as it includes too much area, and changes in threats may occur in regions where the species does not occur with the EOO. However, sampling efforts per species are unknown, hence a need for using EOO as a basis exists.

Following the guidelines (page 58 and after) that state "EOO may be estimated based on ... projected sites of present occurrences [which] ... refers to spatially predicted sites on the basis of habitat maps or models", which in turn "may be derived from interpretation of remote imagery and/or analyses of spatial environmental data ..., or by more formal statistical habitat models ... also referred to as ecological niche models, species distribution models, bioclimatic models and habitat suitability models.", we wanted to **adopt a rather conservative approach** that still fits within the terminology used by the IUCN. Therefore, we opted for determining the EOO by species occurrence records (as we did before) but removing areas that likely do not fit the species climatic niche:

“The minimum convex polygon of species may include large areas where the species is not present, especially for widespread species. For example, *Erythrina velutina* occurs in the seasonally dry tropical biome³² yet a large part of the Amazonian rainforest would be included in its extent of occurrence while it does not occur there. Therefore, fundamentally unsuitable habitat (i.e., water) and climatically unsuitable habitat for each tree species specifically were masked from species’ minimum convex polygon (Fig. 4). This will allow for more accurate quantification of rates of recent threat changes in species’ extent, as we only consider areas where the species can potentially occur.” (lines 440 – 446)

The conservatism of our new method lies in the fact that we do not consider other factors that determine species’ distributions (further explained below) and hence remove less area, hence we potentially overestimate a species’ actual size of habitat. Yet our new method can be considered appropriate, since “it is very rare for detailed and relevant data to be available across the entire range of a taxon. For this reason, the Red List Criteria are designed to incorporate the use of inference, suspicion, and projection, to allow taxa to be assessed in the absence of complete data” (page 19 of the guidelines). And since “larger EOs usually result in a higher degree of risk spreading (and hence a lower overall risk of extinction for the taxon) than smaller EOs, depending on the relevant threats to the taxa.” (page 50 of the guidelines), the conservative estimates of species’ EOO in combination with the choice of stating our rate of recent change values relative to the species EOO makes us believe that our approach can fit the IUCN Red List frame-work.

The choice of using climate zones to determine a species’ suitable habitat is because of multiple reasons:

(1) Making a validated species distribution model per species, as you suggested, would be very challenging within the timeframe of this review and would also outdo the purpose of this study to make an automated, data-driven methodology that can be applied to many taxa and can be updated when new data is available (both for species occurrence records and updated threat maps). Based on the idea that most species distribution models heavily rely on climate, we updated our method by using climate zones to make species’ extent of occurrence more specific, making the extent come closer to a species estimated or projected range or Area Of Habitat and closer in size to a species’ area of occurrence.

(2) We choose not to use ecoregions, also suggested by the reviewer in the next comment, since they are based on species composition and because they make the methodology too restrictive: the species’ extent of occurrence in which we would calculate rates of recent change in different threats would (1) become too specific as only small areas are classified as the same ecoregion, (2) become too sensitive in the way that geolocations of occurrence records are more likely to fall on ecoregion borders (simply because there are more borders) making the estimate of the species range too imprecise, and (3) put too much emphasis on each occurrence record as only a small area around it would be considered suitable habitat for the species. These reasons make that such maps would require an expert to validate the maps, which again outdoes the purpose of this study to make an automated, data-driven methodology.

(3) We did not opt for the use of species’ area of occupancy as the basis of our study, also mentioned by the reviewer, for the same reasons mentioned above, because of the advice against using single grid cell values from globally projected maps as they are less reliable than considering multiple values over a larger areas, and because the AOO is “a scaled metric that represents the area of suitable habitat currently occupied by the taxon” (page 52 of the guidelines) which may underestimate the actual range of a species and hence the threat changes within its area.

(4) The IUCN guidelines suggest another method to ‘remove’ areas from a species’ EOO when it spans uninhabitable regions, namely the use of alpha hulls (page 51 of the guidelines). After some tests with our data, we found this method to bring flaws in the quantification of EOO, as the production of alpha hulls (polygons) for this amount of species (i.e. performing a species-specific check is not ideal) were sometimes mis-shaped.

MAJOR COMMENT 3:

However any estimates of ranges of suitable habitat for species with for example only 5 presence coordinates seem highly unreliable anyway, independent of the method used, so excluding such species from the analysis could be more appropriate, rather than aiming to assess as much as possible species as possible. On the other hand such species may also be the species most in need of a IUCN Red List (re-)assessment, so I understand why the authors have chosen to include as much as possible species. An alternative (although rough) approach to estimate area of occupancy could be to intersect the species' EOO with the terrestrial ecoregions, and to exclude those ecoregions where the species is not recorded, and I am sure there are other possible solutions.

We prefer to include these poorly sampled species for exactly the reason the reviewer points out – they are potentially of greatest concern. However, the rate of change calculations for these species are handled differently in this study and these species are considered separately in the results. Considering our method of using the 95minimum convex polygon to first determine a species' EOO, the proposed alternative for these poorly sampled species is not possible, i.e. the EOO could not be drawn from 5 or less occurrences.

MAJOR COMMENT 4:

My second concern is about the use of occurrence records outside the native range of species. The authors do not seem to mention if such records were included in the convex hull or not, so presumably they used records from around the world for all species? Taking again the example of *Vachellia nilotica*, which has become common in India and Australia (often invasive), and occurs in the Americas too. There are arguments pro and contra including records outside the native range in threat assessments, but if the authors also used records outside the native range, this seems to make the use of the convex hull as basis for the analysis even more problematic (the convex hull of *Vachellia nilotica* would then cover almost the entire tropics worldwide?).

We understand your concern and agree that both including or excluding data points outside a species' native range could be preferred for different reasons. For species where individuals outside the native range are not well established or have small populations, it may be better to exclude non-native areas. However, we believe that including the introduced areas makes the most sense for the goal of this study: In the context of species extinction risk, which is the context of this study, **non-native areas hold value for the persistence of a species** (Lundgren et al., 2023). In addition, including more area in the species' extent makes our method **more conservative**. An extreme example would be the Monterey Pine, *Pinus radiata*, which has a small native range but is well established in various areas around the world (<https://powo.science.kew.org/taxon/urn:lsid:ipni.org:names:77169777-1>).

As controversies exist around using non-native areas, we discuss this point now in the manuscript: "On the other hand, we included occurrence records from the species' native and non-native range, since both could provide opportunities for species' persistence in the context of global change (Dunwiddie & Rogers, 2017)." (lines 426 – 428)

Lundgren, E. J., Wallach, A. D., Svenning, J.-C., Schlaepfer, M., Andersson, A. L. A., & Ramp, D. (2023). Preventing extinction in an age of species migration and planetary change (p. 2023.10.17.562809). bioRxiv. <https://doi.org/10.1101/2023.10.17.562809>

IN-TEXT COMMENTS

Line 256: 'area of occupancy': this term would need to be defined, or does it refer to the same as extent of occurrence in this case (see major comments above)?

The term 'area of occupancy' is dissimilar to the term 'extent of occurrence', which is why we now provide a clear definition in the Methods and removed the term here in the Results section:

“Species with a minimally occupied area, defined here as species with an area of occupancy (i.e., the area within a species’ extent of occurrence which is occupied by that species⁸⁶) smaller than 10 km² defined on a 2 x 2 km grid or species with less than five occurrences, were listed for IUCN Red List assessment prioritization due to their small occupied area (n=9,741).” (lines 429 – 432)

Line 281: ‘area of occurrence’: confusing term as it is a mix between ‘extent of occurrence’ and ‘area of occupancy’

We agree, we adjusted the text to ‘area of occupancy’ as this is how it is stated under the species’ assessment justification (<https://www.iucnredlist.org/species/154179424/157626559#assessment-information>).

Line 437: ‘area of occupancy’: see above

We provide a clear definition now:

“Species with a minimally occupied area, defined here as species with an area of occupancy (i.e., the area within a species’ extent of occurrence which is occupied by that species⁸⁶) smaller than 10 km² defined on a 2 x 2 km grid or species with less than five occurrences, were listed for IUCN Red List assessment prioritization due to their small occupied area (n=9,741).” (lines 429 – 432)

Line 439: ‘small occurrence’: confusing term

We now changed the words to ‘small occupied area’:

“Species with a minimally occupied area, defined here as species with an area of occupancy (i.e., the area within a species’ extent of occurrence which is occupied by that species⁸⁶) smaller than 10 km² defined on a 2 x 2 km grid or species with less than five occurrences, were listed for IUCN Red List assessment prioritization due to their small occupied area (n=9,741).” (lines 429 – 432)

Fig. 3: Could the authors explain why there are no hotspots of built-up expansion in South America and especially Africa (with large expansion of urban areas underway in West-Africa, for example Lagos). Same for deforestation. Not suggesting there has been an error, but this seems counter-intuitive and I would like to understand what would be the reason behind this?

In our study, we identify hotspots (as plotted in figure 3) by overlapping highly exposed species, which are the species that have a rate of recent change value exceeding the 95th percentile threshold for a specific threat. Multiple factors can be the cause for not seeing hotspots in South America. A methodological reason is mentioned in the text already: “Why changes in threat are more easily spotted around the equator can be explained by the unit for threat quantification (percentage of extent), the threshold selection (relative to other species), and the high diversity of small-ranged species at low latitude and fewer larger-ranged species at higher latitudes⁶⁶. Species in other areas could also be affected by threats even when they do not show on these maps. Nevertheless, the identified hotspots are locations with the highest tree species diversity exposed to great change, suggesting these sites and/or species require conservation attention.” (lines 333 – 339) Additionally, the source data for the threats may show a lack of change in the selected time frame (2000-2020).

According to our data source, this is the case for built-up expansion. The two black maps below are copy pasted from our data source (Potapov, P. et al. The Global 2000-2020 Land Cover and Land Use Change Dataset derived from the Landsat Archive: First Results. *Front. Remote Sens.* 3, 856903 (2022)) with the goal to clarify our response. In these maps, it is clear to see that built-up expansion was more extreme in South-East Asia compared to South America, likely causing the lack of hotspots in our results (Figure 3). For deforestation, however, much stronger percentages are seen over the world, suggesting a combination of the two previously described possible causes for not finding a species hotspot: forest loss in South East Asia is stronger compared to South America and tree species

may possibly have a smaller extent of occurrence in those regions. Additionally, as mentioned above, we merely highlight the EOO of species that exceed the 95th percentile of the threats' rate of change, hence white areas do not identify no habitat loss due to the specific threat.

Left: cropped from figure 3 of our manuscript. Right: cropped from figure 5 of Potapov et al. (2022) with the following original caption: Built-up lands and their net change for each $1 \times 1^\circ$ grid cell. **(B)** Built-up lands increase 2000–2020, % cell area.

Left: cropped from figure 3 of our manuscript. Right: cropped from figure 2 of Potapov et al. (2022) with the following original caption: Forest extent, structure, and dynamics for each $1 \times 1^\circ$ grid cell. **(D)** Forest loss, disturbance, and degradation 2000–2020, % year 2000 forest area within a cell.

Line 299 (and elsewhere): 'range': the extent of occurrence should not be used interchangeably with the concept 'range', see major comments above.

As mentioned above, we removed all uses of the word 'range' when referring to an area where a species (potentially) occurs, and we include some examples of how we changed the text. The notion of the word 'range' referred to here was solved by removing part of the sentence. It now reads: "Remarkably, these species comprise more than half (54.2%) of all the tree species assessed here." (lines 282 – 283)

Line 342 (and elsewhere): 'burned areas': I think this should be 'burnt'

The use of the word burn in past tense as an adjective can both be written as 'burnt' and 'burned'. We see why you suggest to change it to burnt here, because we refer to multiple areas. However, instead of changing burned into burnt, we removed the 's' in 'areas'. This makes it consistent with the rest of the text where we use 'burned area'.

Reviewer #2 (Remarks to the Author)

GENERAL FEEDBACK AND MAJOR COMMENT 1:

It is great to see a methodology for flagging current or recent threats to tree species that could be considered when reassessing tree species and could be overlooked.

However, I do not think that the methodology is linked specifically to the IUCN Red List criteria meaning that it is not directly applicable. Criterion B for example requires not only a small extent of occurrence or area of occupancy but also continuing declines and a small number of locations in respect to threat. Criterion A requires declines to be calculated over 3 generations, for trees normally more than 20 years.

We understand your concern. Maybe it was not clear from our manuscript, but the information provided by the rates of recent change in threats to trees is only meant to provide additional information to the experts that do the IUCN assessments. We do not aim to calculate or replace any of the information that is currently used for any of the specific criteria used in the Red List assessment.

We have changed or removed all sentences that may have implied replacing existing criteria, and now focus on how our analyses can be used to flag species for the decadal re-evaluation, e.g.:

“Together with the need for laborious expert-based IUCN Red List re-evaluations every five to ten years³¹, the risks of overlooking pressures especially for rare species in remote areas, suggest that a data-driven systematic approach that quantifies recent changes in threats could aid experts in prioritizing species for new in-depth conservation assessments or re-evaluations³² and therefore could strengthen the critical IUCN Red List assessment work^{27,30-34}.” (lines 95 – 100)

“The threat change quantification on a continuous scale can provide such additional information for experts, not to replace in-depth expert-based assessments but to help prioritize species for these time-consuming assessments despite scarce resources, as our method is transparent and flexible⁷⁹.” (lines 375 – 379)

“Our method can easily be extended to other taxa and can act as an early-warning tool, especially for inconspicuous threats like climate change, allowing for a systematic approach to expedite and broaden IUCN Red List assessments in this time of global change.” (lines 381 – 384)

MAJOR COMMENT 2:

My main concern is with conclusions. **The paper calls for 16,000 "expert-based assessments"**, however, the vast majority of these species have been assessed in the last 10 years with input from experts through the Global Tree Assessments. The majority of the non-evaluated tree species have draft assessments completed or reviewed by experts awaiting publication. If there is no direct link between the results of this analysis and the IUCN Categories and Criteria then this is asking for work to be redone that has already recently been completed by experts. There will of course be a selection of species that have erroneous assessments.

We highly value the work that has been done by experts and we do not request immediate re-evaluations. However, since assessments need to be re-evaluated every 5-10 years which is highly time consuming, our aim is to provide additional information that can help prioritize species for next assessments. By highlighting which species see high rates of change in threats, these species may require prioritization for re-evaluation over species that do not see a change in threats. We provide a method that is transparent and can be applicable to all taxa (with adjustments of threats possibly). We make this clearer now in the text:

“To identify species that require expert-based re-evaluation, we assessed exposure to change in six significant threats over the last two decades for 32,090 tree species” (lines 46 – 47)

“Together with the need for laborious expert-based IUCN Red List re-evaluations every five to ten years³¹, the risks of overlooking pressures especially for rare species in remote areas, suggest that a data-driven systematic approach that quantifies recent changes in threats could aid experts in

prioritizing species for new in-depth conservation assessments or re-evaluations³² and therefore could strengthen the critical IUCN Red List assessment work^{27,30-34}.” (lines 95 – 100)

“We then propose a prioritization for expert-based IUCN Red List re-evaluation based on high exposure to recent changes in threats.” (lines 115 –117)

“Hence, the species prioritization based on rates of climate change can help identify species that require higher priority in the decadal IUCN Red List re-evaluations.” (lines 323 – 325)

“The threat change quantification on a continuous scale can provide such additional information for experts, not to replace in-depth expert-based assessments but to help prioritize species for these time-consuming assessments despite scarce resources, as our method is transparent and flexible⁷⁹. For example, the ~17,000 species exposed to recent global change can be ranked according to the number of threats species are exposed to, or the rate of change of a particular (set of) threat(s), to help identify species most in need of re-evaluation, e.g., considering additional criteria.” (lines 375 – 381)

“For example, the quantification of rates of change in various threats to tree species can inform the experts performing the species-specific assessments on the need for a routine re-assessment.” (lines 355 – 357)

MAJOR COMMENT 3:

The mention of specific species and their assessments in the text is confusing when the results of this study would not impact the results.

It is true that the results of this study do not impact the assessments. The reason why we do include this information, is because the species names together with their IUCN Red List assessment and mentioned threats act as validation or check of how well our analyses compare with what has been said about the species by experts. We now make this explicit in the manuscript:

“These comparisons of species exposed to the fastest rates of change to their expert-based assessments from the IUCN Red List show that the species most exposed to tree cover declines are indeed listed as Endangered due to wood harvesting on the IUCN Red List. These, and other comparisons below, support our method and suggest that species without a Red List status or an outdated status that are exposed to high rates of tree cover decline may likewise currently be threatened and thus are in need of expert-based re-evaluation.” (lines 153 – 158)

MAJOR COMMENT 4:

Least Concern is not no concern, and there are few species that are not seeing global change. For example, the two *Ardisia* species that are flagged in the conclusions, these both have recent IUCN Red List assessments published by the Colombian Plant Specialist Group. Climate change is not flagged in the assessment, but it is unclear what the effects will be to the species. This is part of the reason that climate change is so rarely used in assessments as you note in line 317, as it is hard to estimate the declines for use of criterion A.

We agree that Least Concern does not mean there is no concern. Since we combine multiple IUCN Red List categories into one group, we did not use the category names from the IUCN to avoid confusion. We feel that our terminology of ‘Not Threatened’ incorporates a similar tone, where the species may not be threatened but are also not of no concern. We specify this now clearly in the manuscript. If the reviewer feels that our terminology remains unfit, we are keen to change the terminology for this group of species. The text now reads:

“We assessed the congruence of IUCN Red List conservation status groups with the quantified changes in threats after creating a Threatened group (all species listed as Endangered and more threatened classes), a Vulnerable group (all species listed as Vulnerable), a Not Threatened group (all species listed as Near Threatened or Least Concern, note these species are not of ‘no concern’), and a Data Deficient group (all species listed as Data Deficient).” (lines 198 – 202)

“A large number of these species are potentially being overlooked as being at risk, as 8,119 of the priority candidates are listed as Near Threatened or Least Concern on the IUCN Red List (49.1% of all

species in our Not Threatened conservation status group), 312 are listed as Data Deficient (57.8% of all Data Deficient species), and 5,792 are Not Evaluated (58.4% of all Not Evaluated species). On the other hand, 1,521 of these priority candidates are listed as Endangered or worse on the IUCN Red List (63.9% of all species in our Threatened conservation status group), and 1,649 species are listed as Vulnerable (60.8% of all species in our Vulnerable conservation status group).” (lines 230 – 237)

We further agree that climate change impacts are difficult to estimate for most species. Yet, we make the priority list based on rates of change within the species extent, so that experts are aware of changes and can make an estimated guess whether the species is sensitive to changing climate (e.g. a species only occurs between in areas with rainfall between 500 and 600mm per year). The rate of change values we provide can inform the expert on how much change has been observed over the last 20 years within the species’ extent and whether the species’ extent is close to getting wetter/drier or warmer/colder than the range it currently occurs at. We believe that this kind of information is better than no information or ignoring climate change. Depending on the range of the species and whether it covers e.g. multiple climate zones, a good estimate can be made on whether a species may find habitat in another climate zone or whether it is strictly bound to a specific climate zone. Although no numbers on population declines can be provided at this time, it is helpful to identify species with high rates of climate change to prioritize monitoring and to validate our methods in case a population trend is seen at some point, suggesting that re-evaluation of these species should be done at least every 5-10 years. We include this now in the text:

“While effects of climate change may be difficult to estimate in terms of population trend changes or loss of locations as required to update IUCN Red List statuses, species exposed to high rates of climate change should be re-evaluated regularly in case any effects do become observable. Hence, the species prioritization based on rates of climate change can help identify species that require higher priority in the decadal IUCN Red List re-evaluations.” (lines 320 – 325)

Please, let us know if we misunderstood the reviewer’s comment. With some clarification of the comment, we can make further adjustments to the text.

MAJOR COMMENT 5:

There are many reasons that species can be Data Deficient, including taxonomic. Data Deficient species are those for which an LC and a CR assessment could be equally plausible.

We don’t fully understand the comment regarding Data Deficient species. We do understand that species with this label are deficient in data which is the sole reason they cannot be given an extinction risk status. However, the uncertainty of the status ‘Data Deficient’ can be misleading and can make the species vulnerable (Borgelt et al., 2022). Therefore, we included species with this status in our study, so that they can be included in the prioritization for re-assessments. We are happy to elaborate on any specific comment the reviewer has or make changes to the text if we know where the disagreement with our text lies.

Borgelt, J., Dorber, M., Høiberg, M. A., & Verones, F. (2022). More than half of data deficient species predicted to be threatened by extinction. *Communications biology*, 5(1), 679.

MAJOR COMMENT 6:

It would be interesting to try this analysis on the maps published on the IUCN Red List as these are cleaned maps.

We agree that this would be interesting to try. However, the IUCN only provides range maps for 1956 tree species (<https://www.iucnredlist.org/resources/spatial-data-download>) while we include 32,094 species. The benefit of the IUCN range maps is that they are curated by species experts, yet our species occurrence records have also been heavily cleaned and should represent natural (yet also

non-native) occurrences. To make our species' extent maps more similar to the IUCN species' range maps, we adjusted the methods and remove area with 'unsuitable climate' from the extent, where 'unsuitable' climate is defined by climate zones where the species has no or few (<5% of all) occurrence records.

As a test, we have plotted three different measures of species' extent of occurrence into one map: the IUCN range maps, our previous range maps, and the range maps updated with climate zone information. We did this as an example for 10 randomly chosen species (10 randomly selected letters from the alphabet and the first species having IUCN Red List species' range maps with that letter was selected to show here). It appears that our estimates are in high agreement with the IUCN maps, and hence we did not pursue this further.

Caption for figure below:

10 species with different distribution predictions plotted: Blue lines represent the IUCN species ranges, in red are the 95mcp (minimum convex polygon using 95% of the occurrence records) as used in this paper (used as the EOO in the previous version of the paper), and in grey are the suitable climate zones (described in the manuscript). For the new version of the manuscript, the species extent is defined as the grey area within the red polygon. The black dots represent the species occurrence records used in this study (all occurrence records with labels AAA-C). Note that for *Vernonia cockburniana* and *Kandelia candel* the estimated 95mcp is very small, hence the black dots overlap with the red line, making it invisible.

Neobuxbaumia_mezcalaensis

Cupaniopsis_globosa

Vernonia_cockburniana

Kandelia_candel

IN-TEXT COMMENTS

Line 118 - I do not think this is true. IUCN Red List reevaluations are not prioritised based on likelihood of a different outcome. If there has been a negative change or errors were made, reassessments can happen faster.

We agree. We have reformulated the text, it now reads: "Changing spatial patterns in threats (threat landscapes) are the primary cause for changes in extinction risk¹⁶." (lines 69 – 70)

Line 448 - it is not possible to use assessments conducted under synonymous names unless homotypic, not clear from the text if this included heterotypic synonyms

We have checked our code and found that there was never an incident where the species name was not found on the IUCN Red List but a synonym name of that species was found. Therefore, we removed this line of text from the manuscript.

Line 538 - LR/cd - should be considered NT and not Threatened (this category no longer exists).

Thank you for catching this. We have now changed it in the text and all R scripts, re-analyzed the data and updated the figures and text accordingly. The Methods in the manuscript now reads: "These three groups are: (1) Not threatened, comprising of LC, NT, LR/lc, LR/cd, and LR/nt, (2) Vulnerable, comprising of VU, and (3) Threatened, comprising of EX, EW, RE, CR, and EN." (lines 550 – 551)

REVIEWERS' COMMENTS

Reviewer #1 (Remarks to the Author):

I would like to thank the authors for carefully evaluating my comments and adjusting the analysis and the manuscript accordingly. The paper is now almost suitable for publication in my opinion, although I would appreciate an adjustment in the method to exclude non-suitable areas (see below), and a clarification on non-inclusion of points on the Galapagos islands (see below).

1) The concepts of extent of occurrence and area of occupancy are now defined in the manuscript, making the paper much less confusing. However I'd suggest that the authors to define the concept of AOO when the authors first mention it (line 264). In addition, given the strong focus on IUCN Red Listing, it would be useful if the authors would state more explicitly how the basis of their analysis relates to the concepts of AOO and EOO more explicitly.

2) As the authors state in their response, they now use a geographic area that is somewhere between EOO and AOO. Area of potential occurrence could maybe be a suitable term for this, but maybe the authors prefer to avoid introducing new terms?

3) The intersection of the convex hull with Koppen-Geiger climate zones is a good solution in my opinion. However it still includes whole of the Sahara desert and Saudi Arabia desert for *Vachellia nilotica*, so its estimated exposure to threats will still be much lower than when considering the area it actually occupies. Would excluding desert areas without any tree cover from the species' extents be a solution for this? Maybe through an overlay of the climate zones and the tree cover dataset?

4) About the maps in the response to the reviewers of *Vachellia nilotica* and *Erythrina velutina*: why are the presence points on the Galapagos islands not included? The presence of both species there is well-known. Are the GBIF records there all marked as low quality in the TREECHANGE dataset? On what basis? Too strict data cleaning could have strong consequences on the extent of species used as the basis for analysis?

5) On Line 451 ('Unsuitable habitat for trees was identified by Modis MOD44W for the year 2015'): I'd suggest to make it more explicit that this refers to removing waterbodies (as opposed to climatically unsuitable habitat)

Reviewer #2 (Remarks to the Author):

Thanks for making the changes that I suggested. I think now that you have changed the methodology for calculating the species extent it is clear that there is not a direct link to the IUCN Red List, making the conclusions understandable.

Lines 78-80 are not correct. The document referenced assigns IUCN categories to the tree species, but those assessments are not all on the IUCN Red List at this time.

Reviewer #1 (Remarks to the Author):

I would like to thank the authors for carefully evaluating my comments and adjusting the analysis and the manuscript accordingly. The paper is now almost suitable for publication in my opinion, although I would appreciate an adjustment in the method to exclude non-suitable areas (see below), and a clarification on non-inclusion of points on the Galapagos islands (see below).

Comment 1) The concepts of extent of occurrence and area of occupancy are now defined in the manuscript, making the paper much less confusing. However I'd suggest that the authors to define the concept of AOO when the authors first mention it (line 264). In addition, given the strong focus on IUCN Red Listing, it would be useful if the authors would state more explicitly how the basis of their analysis relates to the concepts of AOO and EOO more explicitly.

Response 1) We wrote out the concept of AOO in line 264 and in line 164. Instead of 'the area of occupancy' we now write it as 'the occupied area within the species' extent of occurrence'.

We also included a line of text at the end of the introduction to state more explicitly how the basis of our analysis relates to the concept of EOO and AOO: "Our approach utilizes species' extent, determined as the minimum convex polygon encompassing 95% of species' high-quality occurrence records, with areas of unsuitable climate and water bodies removed, allowing us to calculate the exposure to threats independent of each species' sensitivity or adaptability to individual threats. In essence, we refined a tree species' extent of occurrence with broadly defined climate niches to remove often vast areas with clearly unsuitable conditions" (original text in black, adjustments in red)

Comment 2) As the authors state in their response, they now use a geographic area that is somewhere between EOO and AOO. Area of potential occurrence could maybe be a suitable term for this, but maybe the authors prefer to avoid introducing new terms?

Response 2) As described above, we use a fine-tuned version of the original term EOO. This area cannot directly be considered as potential occurrence, as such area may also exist outside the species' extent of occurrence. A new term would need to be highly specific and thus require a thorough explanation. Therefore, we like to stay with 'species range', as this more commonly used term together with our thorough explanation of what we specifically mean enhances reader comprehension and facilitates a cohesive flow throughout the paper.

Comment 3) The intersection of the convex hull with Koppen-Geiger climate zones is a good solution in my opinion. However it still includes whole of the Sahara desert and Saudi Arabia desert for *Vachellia nilotica*, so its estimated exposure to threats will still be much lower than when considering the area it actually occupies. Would excluding desert areas without any tree cover from the species' extents be a solution for this? Maybe through an overlay of the climate zones and the tree cover dataset?

Response 3) We agree that considering tree species' area of occupancy could in some cases drastically change estimated exposure to threats, although deserts would often be under low anthropogenic pressure. However, since we do not know all species occurrences in all areas of the world, using species' AOO will strongly underestimate the occupied area of species. Sticking with our fine-tuned version of species' EOO, we indeed may include entire desert areas with the same climate zone whenever occurrences of trees are found there. Actually, many trees occur in the desert, with a

count of 1.8 billion individual trees in the West African Sahara and Sahel (Brandt et al. 2020). Our dataset also includes many tree occurrences in desert areas around the world (Fig.1). These may be linked to oases or other areas with high ground water tables, but in any case some tree species do occur in desert areas making it suboptimal to simply remove these climate zones from our estimation. To make our approach less sensitive to outliers (occurrences that may be located in a desert either as errors or as rare occurrence in a specific hydrological or topoclimatic settings) we use only 95% of the occurrence records per species to identify the extent of occurrence, and additionally, we use only climate zones that cover at least 5% of all grid cells within a 1-km radius buffer around each occurrence.

In addition, we note that some tree species are even linked to this climate zone specifically. One example is the endangered palm species *Medemia argun* (Fig.2), but also other species have occurrences records in the desert. These are a few examples, in decreasing order of number of occurrences in the desert areas plotted in Fig. 3: *Vachellia tortilis* (https://en.wikipedia.org/wiki/Tree_of_T%27%27), *Prosopis farcta*, *Tamarix aphylla*, *Tamarix tetragyna*, *Moringa peregrina*, *Vitex agnus-castus*, *Salix acmophylla*, *Populus euphratica*, *Jatropha spinosa*, *Olea europaea*, *Vachellia nilotica*, and *Grewia tenax*. If we were to remove desert regions from species' EOO, we would likely misrepresent the ranges of these species. Similarly, we refrain from using tree cover data from satellite images as these are highly insensitive to open areas while some tree species may occur only in open regions.

Fig.1 log₁₀ Tree occurrence density in desert areas. The desert and xeric shrubland biome is indicated in grey. The density map of tree occurrence records used in this study was masked by the desert and xeric shrubland biome, hence the yellow to purple color gradient is only applied to these areas.

Fig.2 The Extent of Occurrence of *Medemia argun* is plotted in green. This species native to Egypt and Sudan and is known to only occur in the desert and xeric shrubland biome (<https://en.wikipedia.org/wiki/Medemia>).

Fig.3 The Sahara and some of the Arabic deserts. The dots represent tree species occurrences from 382 species that are located within the desert region plotted here.

Comment 4) About the maps in the response to the reviewers of *Vachellia nilotica* and *Erythrina velutina*: why are the presence points on the **Galapagos islands** not included? The presence of both species there is well-known. Are the GBIF records there all marked as low quality in the TREECHANGE dataset? On what basis? Too strict data cleaning could have strong consequences on the extent of species used as the basis for analysis?

Response 4) Both species indeed have occurrence records on the Galapagos islands, and they are of good enough quality. Hence they are included in our dataset. The reason why they are not included in the figures in the response to the reviewers is that these figures show the extent of occurrence created with a 95% minimum convex polygon (mcp). This means that the extent of occurrence is drawn around 95% of the occurrence records that leads to the smallest polygon. In the instances of the two example species, the records on the islands were thereby excluded from the polygon. We adopted this method to reduce the risk of possible remaining outliers having undue effects on our range estimates, making our approach more conservative. Additionally, this effect (the use of 95% of the data compared to 100%) will likely mainly affect large-ranged species where any underestimation of the actual extent is unlikely to impact the exposure to risk much.

Comment 5) On Line 451 ('Unsuitable habitat for trees was identified by Modis MOD44W for the year 2015'): I'd suggest to make it more explicit that this refers to removing waterbodies (as opposed to climatically unsuitable habitat)

Response 5) We adjusted the original line 451 to "Waterbodies were identified by Modis MOD44W for the year 2015⁹⁰" (original text in black, adjustments in red)

We additionally removed other instances of 'unsuitable habitat'.

Reviewer #2 (Remarks to the Author):

Thanks for making the changes that I suggested. I think now that you have changed the methodology for calculating the species extent it is clear that there is not a direct link to the IUCN Red List, making the conclusions understandable.

Comment 1) Lines 78-80 are not correct. The document referenced assigns IUCN categories to the tree species, but those assessments are not all on the IUCN Red List at this time.

Response 1) We clarified this point in the text: "With the latest Global Tree Assessment (GTA), 92.7% of all 57,922 tree species²³ have been assigned a conservation status that will be included on the International Union for the Conservation of Nature (IUCN) Red List of Threatened Species^{5,24} (hereafter 'IUCN Red List')." (original text in black, adjustments in red)

References

Brandt, M., Tucker, C.J., Kariryaa, A. *et al.* An unexpectedly large count of trees in the West African Sahara and Sahel. *Nature* **587**, 78–82 (2020). <https://doi.org/10.1038/s41586-020-2824-5>